# Differential roles of Na$_V$1.2 and Na$_V$1.6 in neocortical pyramidal cell excitability

Joshua D Garcia[1,2]*, Chenyu Wang[1,2], Ryan PD Alexander[1,2], Emmie Banks[3], Timothy Fenton[4], Jean-Marc DeKeyser[5], Tatiana V Abramova[5], Alfred L George Jr[5], Roy Ben-Shalom[4], David H Hackos[3], Kevin J Bender[1,2]*

[1]Weill Institute for Neurosciences, University of California, San Francisco, San Francisco, United States; [2]Department of Neurology, University of California, San Francisco, San Francisco, United States; [3]Department of Neuroscience, Genentech, Inc., South San Francisco, United States; [4]Department of Neurology, MIND Institute, University of California Davis School of Medicine, Sacramento, United States; [5]Department of Pharmacology, Northwestern University Feinberg School of Medicine, Chicago, United States

**\*For correspondence:**
jgarc4@gmail.com (JDG);
kevin.bender@ucsf.edu (KJB)

## eLife Assessment

This manuscript presents a clever and powerful approach to examining differential roles of Nav1.2 and Nav1.6 channels in excitability of neocortical pyramidal neurons, by engineering mice in which a sulfonamide inhibitor of both channels has reduced affinity for one or the other channels. Overall, the results in the manuscript are **compelling** and give **important** information about differential roles of Nav1.6 and Nav1.2 channels. Activity-dependent inactivation of NaV1.6 was also found to attenuate seizure-like activity in cells, demonstrating the promise of activity-dependent NaV1.6-specific pharmacotherapy for epilepsy.

**Abstract** Mature neocortical pyramidal cells functionally express two sodium channel (Na$_V$) isoforms: Na$_V$1.2 and Na$_V$1.6. These isoforms are differentially localized to pyramidal cell compartments, and as such are thought to contribute to different aspects of neuronal excitability. But determining their precise roles in pyramidal cell excitability has been hampered by a lack of tools that allow for selective, acute block of each isoform individually. Here, we leveraged aryl sulfonamide-based molecule (ASC) inhibitors of Na$_V$ channels that exhibit state-dependent block of both Na$_V$1.2 and Na$_V$1.6, along with knock-in mice with changes in Na$_V$1.2 or Na$_V$1.6 structure that prevents ASC binding. This allowed for acute, potent, and reversible block of individual isoforms that permitted dissection of the unique contributions of Na$_V$1.2 and Na$_V$1.6 in pyramidal cell excitability. Remarkably, block of each isoform had contrasting—and in some situations, opposing—effects on neuronal action potential output, with Na$_V$1.6 block decreasing and Na$_V$1.2 block increasing output. Thus, Na$_V$ isoforms have unique roles in regulating different aspects of pyramidal cell excitability, and our work may help guide the development of therapeutics designed to temper hyperexcitability through selective Na$_V$ isoform blockade.

## Introduction

Voltage-gated sodium channels (Na$_V$) are critical for all aspects of neuronal excitability, from action potential (AP) initiation to axonal propagation, transmitter release, and dendritic excitability (*Meeks and Mennerick, 2007*; *Shu et al., 2007*; *Palmer and Stuart, 2006*; *Spratt et al., 2021*; *Spratt et al., 2019*; *Nelson et al., 2024*). In mature neocortical pyramidal cells, electrogenesis and subsequent

propagation of APs is systematically regulated with $Na_V$ recruitment first initiated in the axon initial segment (AIS), with forward propagation along the axon and backpropagation into soma and dendrites (*Shu et al., 2007*; *Palmer and Stuart, 2006*; *Baranauskas et al., 2013*). This coordinated process is supported by membrane expression of the $Na_V$ isoforms $Na_V1.6$ and $Na_V1.2$. Current models, based on immunostaining of channels and empirical measurements of excitability, suggest that $Na_V1.2$ and $Na_V1.6$ are differentially expressed across neurites. In the AIS, $Na_V1.6$ predominates, with the highest membrane density in the AP initiation region of the AIS most distal to the soma. $Na_V1.2$, by contrast, has higher relative membrane density in the proximal AIS (*Shu et al., 2007*; *Hu et al., 2009*; *Tian et al., 2014*; *Ye et al., 2018*; *Yu et al., 2008*; *Jenkins and Bender, 2025*). Along the axon, $Na_V1.6$ appears enriched at nodes of Ranvier and terminals (*Caldwell et al., 2000*; *Kaplan et al., 2001*). Somatodendritic densities are far lower, and current models suggest that the somatic and perisomatic membrane expresses equal levels of $Na_V1.6$ and $Na_V1.2$, whereas dendritic regions more distal to the soma are enriched exclusively with $Na_V1.2$ (*Spratt et al., 2021*; *Spratt et al., 2019*; *Nelson et al., 2024*; *Fleidervish et al., 2010*). Thus, each channel isoform likely has a unique role in AP initiation and propagation simply based on their differential distribution in neuronal compartments.

To date, efforts to understand the unique contributions of different $Na_V$ isoforms have relied largely on experiments in which channel genetic expression has been manipulated, either through constitutive or conditional knockout approaches. While these approaches have strong merit, interpretation is complicated by compensatory changes in other $Na_V$ isoforms or other ion channel classes (*Spratt et al., 2021*; *Spratt et al., 2019*; *Katz et al., 2018*; *Zhang et al., 2021*). Pharmacological approaches, which are both acute and reversible, would be preferred, but identifying highly selective compounds that target particular $Na_V$ isoforms is difficult, as *ScnXa* gene family isoforms have high degrees of amino acid sequence homology. Some compounds exhibit high potency, but differences in the half-maximal inhibitory concentration ($IC_{50}$) for individual isoforms can be limited (*Ragsdale and Avoli, 1998*; *Denomme et al., 2020*; *Bosmans et al., 2006*; *Filipis et al., 2023*). This is especially true for the separation of $Na_V1.2$ from $Na_V1.6$, as these two channels have especially high sequence similarity (*Catterall et al., 2005*; *de Lera Ruiz and Kraus, 2015*).

Aryl sulfonamide compounds (ASCs) constitute a unique class of $Na_V$ inhibitors that potently bind activated $Na_V$ isoforms, resulting in a stabilization of the inactivated state (*Ahuja et al., 2015*; *Goodchild et al., 2024*; *Johnson et al., 2024*; *Johnson et al., 2022*; *Roecker et al., 2017*). The binding pocket is shielded from ASCs when channels are closed due to the positioning of the S4 voltage sensing domain, but is bound rapidly by ASMs when exposed. Unbinding is promoted by strong hyperpolarization, but occurs much more slowly at physiological voltages (e.g., tau of $10^2$–$10^3$ s at –80 and –40 mV, respectively) (*Ahuja et al., 2015*). In spiking neurons, this essentially imparts use dependence to ASC block (*Ahuja et al., 2015*), as the probability of channel block will increase in proportion with AP rates.

ASCs exhibit high potency for a subset of $Na_V$ isoforms: $Na_V1.2$, $Na_V1.6$, and $Na_V1.7$ (*Ahuja et al., 2015*; *Goodchild et al., 2024*; *Johnson et al., 2024*; *Johnson et al., 2022*). These isoforms share sequence homology at the binding pocket, each containing a tyrosine–tryptophan (YW) motif that helps stabilize ASC binding. By contrast, $Na_V1.1$ and $Na_V1.3$ harbor a serine–arginine (SR) sequence at the same site. Mutagenesis of $Na_V1.7$ from YW to SR results in a 145-fold decrease in ASC binding without affecting channel biophysical properties (*Ahuja et al., 2015*). Here, we engaged a similar strategy for $Na_V1.2$ and $Na_V1.6$, using knock-in mice with YW→SR substitutions in either or both channels. This allows for selective, potent ASC-mediated block of YW-containing channels while preserving the function of SR-containing channels. With these tools, we dissected the individual roles of $Na_V1.2$ and $Na_V1.6$ in neocortical pyramidal cell excitability. We found that $Na_V1.2$ and $Na_V1.6$ have unique—and at times conflicting—effects on overall AP excitability. Moreover, results suggest that ASCs can function as 'on-demand' pharmacological inhibitors that help normalize cellular excitability in seizure-like conditions. Together, this work highlights the importance of distinct $Na_V$ isoforms localized to specific cellular domains and their respective contribution to AP properties to help facilitate activity across pyramidal neurons.

# Methods

**Key resources table**

| Reagent type (species) or resource | Designation | Source or reference | Identifiers | Additional information |
|---|---|---|---|---|
| Gene (*Homo sapiens*) | *Scn2a* Human tagged ORF clone | Origene | NM_021007 | Na$_V$1.2 |
| Gene (*Homo sapiens*) | *Scn8a* Human tagged ORF clone | Origene | NM_014191 | Na$_V$1.6 |
| Strain, strain background (*Mus musculus*, both sexes) | C57BL/6J | Jackson Laboratory | RRID:IMSR_JAX:000664 | Wildtype mice |
| Strain, strain background (*Mus musculus*, both sexes) | C57BL/6N | Janvier Labs | RRID:IMSR_RJ:C57BL-6NRJ | Mating with founders |
| Strain, strain background (*Mus musculus*, both sexes) | *Scn8a* YW→SR KI | Genentech; **Deng et al., 2023**; PMID:37352856 | N/A | Na$_V$1.6 GNE-4076 insensitive mice |
| Strain, strain background (*Mus musculus*, both sexes) | *Scn2a* YW→SR KI | Genentech; this manuscript | N/A | Na$_V$1.2 GNE-4076 insensitive mice |
| Strain, strain background (*Mus musculus*, both sexes) | Dual *Scn8a/2a* YW→SR KI | Genentech; this manuscript | N/A | Both GNE-4076 insensitive |
| Cell line (*Homo sapiens*) | HEK293T | ATCC CRL-3216 | RRID:CVCL_0063 | IC$_{50}$ curves; biophysics |
| Cell line (*Mus musculus*) | ND7/23 (low Na$_V$) | Al George Lab; **Vanoye et al., 2024** | PMID:38771640 | IC$_{50}$ curves; biophysics |
| Recombinant DNA reagent | pIR-CMV-SCN2A-Variant-1-IRES-mScarlet | Addgene | RRID:Addgene_162279 | Na$_V$1.2 Plasmid #162279 |
| Recombinant DNA reagent | pcDNA4TO-SCN8A-Variant-3-IRES-mScarlet | Addgene | RRID:Addgene_209411 | Na$_V$1.6 Plasmid #209411 |
| Sequence-based reagent | 2A_Y1564S_W1565R F | This paper | PCR primers | ATTCTGTCCCGGATTAATCTGGTGTTTATTGTTCT |
| Sequence-based reagent | 2A_Y1564S_W1565R R | This paper | PCR primers | AGATTAATCCGGGACAGAATGTTTGTCATTTCTTGA |
| Sequence-based reagent | 8A_Y1555S_W1556R F | This paper | PCR primers | CATCCTCTCCCGGATTAACCTGGTGTTTGTT |
| Sequence-based reagent | 8A_Y1555S_W1556R R | This paper | PCR primers | GGTTAATCCGGGAGAGGATGTTCTCCATCTG |
| Chemical compound, drug | GNE-4076 or Compound 5 | Merck & Co, Inc; **Roecker et al., 2017** | PMID:28389149 | Used at 200 nM |
| Software, algorithm | IGOR Pro v6.3 & v9 | Wavemetrics | RRID:SCR_000325 | N/A |
| Software, algorithm | Prism 10 | GraphPad | RRID:SCR_002798 | N/A |
| Software, algorithm | NEURON Compartmental Models | modelDB | RRID:SCR_007271 and SCR_003105 | https://modeldb.science/2019342 |
| Software, algorithm | Python | https://www.python.org/ | RRID:SCR_008394 | N/A |
| Software, algorithm | Benchling | https://www.benchling.com/ | RRID:SCR_013955 | N/A |

## Experimental models and subjects details

### Mouse strains

All animal procedures are in accordance with the Institutional Animal Care and Use Committee (IACUC) guidelines in accordance with the University of California, San Francisco (UCSF). The following mouse strains were used in this study: C57BL/6J, YW→SR Na$_V$1.2 KI (*Scn2a*$^{SR/SR}$), YW→SR Na$_V$1.6 KI (*Scn8a*$^{SR/SR}$), and YW→SR dual KI (*Scn8a/2a*$^{SR/SR}$). All experimental procedures were performed on mice maintained in-house on a 12:12 hr light–dark cycle under standard conditions with ad libitum access to food and water. For genotyping, genomic DNA was isolated from tail clip biopsies for PCR. Both male and female mice aged postnatal day (P)18–59 were used across all genotypes. C57BL/6J mice were obtained from Jackson Laboratories, and YW→SR KI mice were developed by the Hackos Lab (Genentech).

### Generation of *SCN8A* or *SCN2A* YW→SR KI mouse

As described previously (*Deng et al., 2023*), CRISPR/Cas9 technology (*Cong et al., 2013*; *Mali et al., 2013*) was used to generate a genetically modified mouse strain with either an *Scn8a* or *Scn2a* YW→SR knock-in mutation. A single-guide RNA (sgRNA) target and protospacer adjacent motifs (PAM) were identified for *Scn8a* ENSMUSG00000023033 or *Scn2a* ENSMUSG00000075318 genomic regions of interest using the CRISPR design tool (Benchling) that uses the algorithm described by *Hsu et al., 2013* to provide 'MIT' specificity scores for each sgRNA, as well as the top 15 predicted off-target loci and corresponding MIT off-target scores. The same guide target and PAM were used for both genes. Guide target: 5′ CATTCTCTACTGGATTAATC 3′; PAM: TGG with an algorithm score of 42.3.

For the YW→SR mutation on the *Scn8a* gene, predicted cut sites are between 100,933,463–100,933,464 genome coordinates. The following oligonucleotide donor sequence was used: 5′ ATGCTTATCTGCCTTAACATGGTGACCATGATGGTGGAGACAGACACACAGAGCAAGCAGATGG AGAACATTCTCTCTCGGATTAATCTGGTCTTCGTCATCTTCTTCACCTGCGAGTGTGTGCTCAAAATG TTTGCCTTGAGACACTACTATTTC 3′. The first point mutation of Y1553S (TAC→TCT) is located at 100,933,454–100,933,456 genome coordinates, and a second point mutation of W1554R (TGG→CGG) is located at 100,933,457–100,933,459 genome coordinates.

For the YW→SR mutation on the *Scn2a* gene, predicted cut sites are between 166,900 and 166,901 genome coordinates. The following oligonucleotide donor sequence was used: 5′ GAAATAGTAGTG TCTCAAGGCAAACATTTTGAGCACACACTCGCAGGTGAAGAAGATGACGAAGACCAGGTTGATC CGAGAGAGAATGTTCTCCATCTGCTTGCTCTGTGTGTCTGTCTCCACCATCATGGTCACCATGT TAAGGCAGATAAGCAT 3′. The first point mutation of Y1553S (TAC→TCT) is located at 101,035,573–101,035,575 genome coordinates, and a second point mutation of W1554R (TGG→CGG) is located at 101,035,576–101,035,578 genome coordinates. Additionally, two silent mutations were created in the beginning of the gRNA to prevent Cas9 from cutting the donor oligo. The first silent mutation (ATT→ATC) is located at 101,035,579–101,035,581 genome coordinates, and the second silent mutation (AAT→AAC) 101,035,582–101,035,584 genome coordinates.

After homology-directed repair of Cas9-induced chromosome breaks with the oligonucleotide donor, the YW→SR protein will be expressed. Once an sgRNA decision was finalized, the off-target list was used to identify the top 15, and next-generation sequencing (NGS) amplicon primers were designed for the on-target locus, and each of the off-target synthetic guide RNAs was obtained from Synthego. CAS9 protein was obtained from PROTEIN SOURCE and complexed with sgRNA before microinjection. Reagent concentrations for microinjection were as follows: 25 ng/μl Cas9 mRNA (Thermo Fisher; A29378) + 13 ng/μl sgRNA (Synthego), Oligonucleotide donor (50 ng/μl) (IDT).

After zygote microinjection and embryo transfer, genomic DNA was prepared from tail tip biopsies of potential G0 founders, and G0 animals were first analyzed by droplet digital PCR (*Hindson et al., 2011*) (Bio-Rad). Primers were used to amplify the HDR event (ON Target) and the 15 most likely off-target sites. Only G0 mosaic founders positive for the intended mutation were screened by targeted amplicon NGS. Amplicons were submitted for NGS analysis.

Founders were selected for mating with wildtype C57BL/6N mice for germline transmission of the gene-edited chromosome. Subsequent analysis of genomic DNA from G1 pups was used to confirm germline transmission of the targeted gene and the absence of off-target hits elsewhere in the genome.

**Table 1.** Mutagenic primer sequences for Na$_V$1.2 and Na$_V$1.6.

| Primer name | Sequence (5′ to 3′) |
|---|---|
| 2A_Y1564S_W1565R_F | ATTCTGTCCCGGATTAATCTGGTGTTTATTGTTCT |
| 2A_Y1564S_W1565R_R | AGATTAATCCGGGACAGAATGTTTGTCATTTCTTGA |
| 8A_Y1555S_W1556R_F | CATCCTCTCCCGGATTAACCTGGTGTTTGTT |
| 8A_Y1555S_W1556R_R | GGTTAATCCGGGAGAGGATGTTCTCCATCTG |

To generate the dual *Scn8a/2a$^{SR/SR}$* line, *Scn2a$^{SR/SR}$*, and *Scn8a$^{SR/SR}$* mice were crossed, and analysis of genomic DNA was used to confirm germline transmission of the targeted gene.

## Method details
### Synthesis of GNE4076
GNE-4076 was synthesized as previously described in *Roecker et al., 2017*, and is Compound 5 in the original manuscript. We use GNE-4076 throughout this manuscript.

### Generation of *SCN2A* YW→SR and *SCN8A* YW→SR constructs
The double mutation Y1564S/W1565R was introduced into the adult splice isoform of recombinant human Na$_V$1.2 (NCBI accession number NM_021007; AddGene #162279) using site-directed mutagenesis as previously described (*Thompson et al., 2023*). The corresponding mutations (Y1555S/W15556R) were engineered in recombinant human Na$_V$1.6 (adult isoform) as previously described (*Vanoye et al., 2024*). Mutagenic primer sequences are presented in *Table 1*. All plasmids were nanopore sequenced (Primoridium Labs, Arcadia, CA) to confirm the variants and exclude unwanted mutations.

### Electrophysiology in immortalized cell lines
HEK293T cells were transiently transfected with WT or mutant Na$_V$1.2 using the Invitrogen Lipofectamine LTX kit, whereas Na$_V$1.6 plasmids were electroporated into the LoNa$_V$ derivative of ND7/23 cells using Maxcyte technology as previously described (*Vanoye et al., 2024*). HEK293T cells were acquired from ATCC (CRL-3216) where mycoplasma testing shows no detection of contamination. LoNa$_V$ derivative of ND7/23 cells is a mix of rat dorsal root ganglion neurons and mouse neuroblastomas that have been modified and previously validated by Al George's Lab as described in *Vanoye et al., 2024*. These cells were not generated from the original source of the parental line ND7/23; therefore, we are unable to authenticate with ATCC. LoNa$_V$ ND7/23 cells have been tested for mycoplasma and show no detection of contamination.

Na$_V$ channel currents were recorded from HEK293 or ND7/23 cells via whole-cell patch clamp using a Molecular Devices Axopatch 200B amplifier. The recording pipet intracellular solution contained (in mM): 120 CsF, 10 NaCl, 2 MgCl$_2$, 10 HEPES, adjusted to pH 7.2 with CsOH. The extracellular recording solution contained (in mM): 155 NaCl, 3 KCl, 1 MgCl$_2$, 1.5 CaCl$_2$, 10 HEPES, adjusted to pH 7.4 with NaOH. Currents were recorded at a 20-kHz sampling frequency and filtered at 5 kHz. Series resistance compensation was applied at 80%. Solutions containing GNE-4076 were applied using a Fluicell Dynaflow perfusion system.

We characterized dose–response curves in both WT and mutant Na$_V$1.2 and Na$_V$1.6 channels by pulsing cells from –80 to –12 mV for 20 ms at a rate of 0.5 Hz to establish a baseline current. Cells were then perfused with GNE-4076 starting at 30 nM and increased up to 100 µM. To allow adequate GNE-4076 onboarding to NaV isoforms/mutants at each successive dose, cells were depolarized to –12 mV for 10 s followed by similar test pulses used to acquire baseline current. Between dose increases, cells were held at –80 mV to allow adequate unbinding of GNE-4076.

We also characterized the activation and steady-state inactivation properties of both WT and mutant Na$_V$1.2 and Na$_V$1.6 channels. To measure activation, we used a holding voltage of –80 mV, a short 20ms pre-pulse to –120 mV, and a 30-ms pulse to voltages ranging from –100 to –20 mV in steps of 5 mV at a rate of once per 3 s. P/4 leak subtraction was used to reduce leak currents. The peaks of the resulting Na$_V$ currents were measured. Conductance was calculated using the equation: *G = I/*

($V - V_{Na}$), normalized, and plotted as a function of voltage. To measure inactivation, we started at a holding voltage of –120 mV to bring the channels fully into the closed state (about 1 min). We then pulsed to –12 mV while reducing the holding voltage from –120 to +30 mV in steps of 5 mV at a rate of once per 5 s. The peaks of the resulting $Na_V$ currents were measured, normalized, and plotted as a function of voltage. To quantify these biophysical properties, we fit the activation and inactivation curves to the following Boltzmann equations:

$$G/Gmax = 1 - \frac{1}{1 + e^{(v - v_{1/2}/sf)}} \text{ (activation) } I/Imax = \frac{1}{1 + e^{(v - v_{1/2}/sf)}} \text{ (inactivation)}$$

$V_{1/2}$ is the voltage where half-maximal peak conductance (or current) was observed, and $sf$ is the slope factor.

## Ex vivo electrophysiology

Mice aged P18–P59 were anesthetized with isoflurane prior to harvesting the brain. Dissected brains were immediately placed in cutting solution (4°C) containing 87 mM NaCl, 25 mM NaHCO$_3$, 25 mM glucose, 75 mM sucrose, 2.5 mM KCl, 1.25 mM NaH$_2$PO$_4$, 0.5 mM CaCl$_2$, and 7 mM MgCl$_2$ that is bubbled with 5% CO$_2$/95% O$_2$. Coronal slices were prepared from the medial prefrontal cortex (PFC) at a thickness of 250 μm and placed in a holding chamber warmed to 33°C for 30 min. Slices were then allowed to recover at room temperature until recording. Recordings were performed at 31–33°C in a solution containing 125 mM NaCl, 2.5 mM KCl, 2 mM CaCl$_2$, 1 mM MgCl$_2$, 25 mM NaHCO$_3$, 1.25 mM NaH$_2$PO$_4$, and 25 mM glucose that is bubbled with 5% CO$_2$/95% O$_2$. Osmolarity of the recording solution was adjusted to approximately 309 mOsm.

Neurons were identified using differential interference contrast optics for conventional visually guided whole-cell recordings. For current-clamp experiments, pipettes were pulled from Schott 8250 glass, with a tip resistance of 3–4 MΩ, and filled with a K-gluconate-based internal solution containing 113 mM K-gluconate, 9 mM HEPES, 4.5 mM MgCl$_2$, 0.1 mM EGTA, 14 mM Tris$_2$-phosphocreatine, 4 mM Na$_2$-ATP, 0.3 mM Tris-GTP; 290 mOsm; pH 7.2–7.25. All data were corrected for measured junction potentials of 12 mV.

All electrophysiology data were acquired through custom protocols generated in IgorPro (Wavemetrics) via Multiclamp 700A or 700B amplifiers (Molecular Devices). AP waveform measurements were acquired at 50 kHz and low-pass Bessel filtered at 20 kHz for all experiments except for data presented in Figure 6 (acquired at 10 kHz and filtered at 3 kHz). Pipette capacitance was compensated in all current-clamp recordings to 50% of the fast capacitance measured after membrane seals were established in voltage clamp. The bridge was balanced, and series resistance was kept <18 MΩ in all recordings. Any cells with input resistance changes exceeding ±15% were omitted from the dataset. A quartz electrode holder (Sutter Instrument) was used to collect all recordings to minimize physical drift of recording electrodes.

All recordings were made from medial PFC layer 5b, thick-tufted pyramidal tract (PT) neurons. Pyramidal neuron identity was confirmed by assessing membrane responses to hyperpolarizing current (–400 pA, 120 ms), with PT neurons defined as those that exhibit membrane depolarization overshoot that peaks within 90 ms of current step offset (*Clarkson et al., 2017*). AP threshold, AIS d$V$/d$t$ and peak d$V$/d$t$ measurements were determined from either the first AP or all APs within a spike train evoked by a stimulus. AP threshold was defined as the membrane potential ($V_m$) when the AP velocity (d$V$/d$t$) exceeds 15 V/s. Somatic peak (V/s) was defined as the max value in the first derivative (d$V$/d$t$) of an AP. AIS max (V/s) was defined as the trough or saddle point in the second derivative (d$^2V$/d$t^2$) that occurs during the depolarizing phase of an AP before the somatic peak (*Favero et al., 2018*). AIS inflection point was defined as the peak of the second derivative (d$^2V$/d$t^2$) that occurs prior to the AIS max or somatic peak. Frequency was defined as the spike number in a train elicited per second (spikes/s or Hz). Afterhyperpolarization was defined as the minimum voltage between two spikes in a train.

Spike trains were evoked by injecting current (250–350 pA, 10 s or 300 ms duration) in the presence or absence of 200 nM GNE-4076. In recovery experiments, neurons were held at –12 mV for 30 s (voltage clamp) to depolarize cells, activate $Na_V$ channels, and expose the ASC binding pocket to maximize GNE-4076 onboarding and $Na_V$ inhibition. The interstimulus interval given during the recovery period was as follows: 2 s, ~20–30 s duration; 5 s, ~30–45 s duration; 15 s, ~60–90 s

duration; 30 s, ~120–180 s duration; and 60 s, ~60–240 s duration. In experiments mimicking cellular activity, a post-synaptic potential train was randomly generated using a Poisson probability distribution function for 60 s with a frequency of 50 Hz and amplitude standard deviation of 200 pA.

For nucleated patch voltage-clamp experiments, recordings were made at 31–33°C in a solution containing 125 mM NaCl, 2.5 mM KCl, 2 mM $CaCl_2$, 1 mM $MgCl_2$, 25 mM $NaHCO_3$, 1.25 mM $NaH_2PO_4$, 15 mM glucose, 4 mM TEA, 1 mM 4-AP, and 10 µM nifedipine, with or without 200 nM GNE-4076, (bubbled with 5% $CO_2$/95% $O_2$). A Cs-methanesulfonate-based recording solution was used: 110 mM $CsMeSO_3$, 40 mM HEPES, 1 mM KCl, 4 mM NaCl, 4 mM Mg-ATP, 10 mM Na-phosphocreatine, 0.4 mM $Na_2$-GTP, and 0.1 mM EGTA; 290 mOsm; pH 7.2–7.25. Whole-cell recordings were established before withdrawing pipettes from the slice, pulling a region of the somatic membrane with the pipette. Neurons were held to –80 mV and stepped to –12 mV every 2 s, 5–10 times. Then neurons were held to –12 mV for 10 s to onboard GNE-4076, then returned to –80 mV and probed for $Na_V$ recovery with steps to –12 mV every 2 s. Leak currents were subtracted with a P/8 protocol using steps from –80 to –90 mV. All data were corrected for measured junction potentials of 12 mV.

## Compartmental modeling

A compartmental model was constructed within the NEURON environment to simulate a layer 5 pyramidal neuron as described before (*Spratt et al., 2021*; *Hallermann et al., 2012*; *Quinn et al., 2024*). This model is available at ModelDB (2019342, https://modeldb.science/2019342). A multi-compartment model, originally developed by the Blue Brain Project, was implemented to reflect the morphology detailed by *Ramaswamy and Markram, 2015* and electrophysiological features were adjusted to reflect empirically obtained data (*Spratt et al., 2021*; *Hallermann et al., 2012*). We modified the model by replacing the aggregated sodium conductances (NaT and NaP) with distinct $Na_V$1.2 and $Na_V$1.6 channels to match empirically observed distributions (*Hu et al., 2009*). $Na_V$1.2 and $Na_V$1.6 channels were distributed throughout the cell with equal levels in the soma and 20 µm of the proximal dendrites. $Na_V$1.2 was solely expressed in dendrites more distal to the soma, inferred based on AP-evoked sodium imaging observations in $Scn2a^{+/-}$ conditions (*Nelson et al., 2024*). The two axonal compartments were subdivided into an AIS and distal axon. Within the AIS, $Na_V$1.2 and $Na_V$1.6 were distributed with increased $Na_V$1.2 in the proximal AIS and increased $Na_V$1.6 in the distal AIS to recapitulate the channel distribution as previously observed empirically (*Hu et al., 2009*). $Na_V$1.2 was not included in the distal AIS or axon where only $Na_V$1.6 is present, including an enriched region to model a node of Ranvier. To simulate blocking of $Na_V$1.2 and $Na_V$1.6, each channel's density was globally reduced in 10% increments from 100% to 0%. Both $Na_V$1.2 and $Na_V$1.6 channels were represented using the Hodgkin-Huxley formalism (*Hodgkin and Huxley, 1952*). Parameter optimization for both channels was conducted using an evolutionary algorithm from BluePyOpt (*Van Geit et al., 2016*) and adapted for use with the computational resources at the National Energy Research Computing Center similar to *Ladd et al., 2022*.

To validate parameter sensitivity, we constructed multiple variations of our model with different distributions of $Na_V$1.2 and $Na_V$1.6 in the AIS. The crossover point at which the $Na_V$1.2 and $Na_V$1.6 distribution curves intersect was shifted distally in increments of 3.75 µm to make five variations of the AIS with increasing density of $Na_V$1.2. With the wildtype crossover point located 15 µm distal to the soma, the five right-shifted variations have crossover points at 18.75, 22.5, 26.25, 30, and 33.75 µm, respectively. Two variations of the AIS with increased $Na_V$1.6 density were created by shifting the crossover point proximally in 3.75 µm increments to 11.25 and 7.5 µm. To see how our model would behave with different AIS sodium channel distributions, we varied the ratio of $Na_V$1.2: $Na_V$1.6 density in the whole cell to test different conditions at each AIS crossover point and measured the resulting threshold, AIS peak, and somatic peak of the phase plane plot. $Na_V$1.2 percentage was decreased from 90% to 0% in 10% increments as $Na_V$1.6 percentage was concurrently increased from 10% to 100% in 10% increments. The $Na_V$1.2: $Na_V$1.6 ratio of 100%:0% did not produce any APs and was therefore not included in the analysis. In instances where the somatic portion of the phase plane was not a true peak that consisted of adjacent values less than a local maxima, the prominence was estimated by taking the index of the second derivative value closest to zero and calculating the d$V$/d$t$ value at that index.

## Quantification and statistical analysis

Data are reported as absolute values or the absolute difference from baseline (delta, Δ). For Δ values, data were normalized either to the initial 500 ms of the stimulus (Figures 2 and 3) or baseline spiking (Figures 4 and 5). Time course graphs are represented as a mean ± standard error. Summary graphs are represented with box plots showing the median, quartiles, and 90% tails or with violin plots. All summary graphs overlay individual datapoints and represent recordings from single cells (reported *n*) for all electrophysiology experiments. Data were obtained from 5 to 9 animals (both sexes) per condition, which are standard group sample sizes used in the field. Group sizes were determined based on sample sizes previously used for similar studies. Power analysis was not required for this study. No attrition applies to our study, and all data collected is included. Analysis was performed blind to genotype ± drug. Statistical analysis was performed using Prism 10 (GraphPad Software). Quantified mean ± standard error and statistical test used is noted in figure legends. Significance was set at an alpha value of 0.05, and 'ns' indicates no significance.

## Results

### Aryl sulfonamides selectively bind and inhibit $Na_V$ isoforms containing the YW motif

ASCs exhibit high affinity for an extracellular region within voltage sensing domain IV (VSD IV) of select $Na_V$ channels with a conserved tyrosine–tryptophan (YW) motif (*Ahuja et al., 2015*; *Figure 1A and B*). This YW motif found on $Na_V$1.2, $Na_V$1.6, and $Na_V$1.7 allows for ASC stabilization following channel inactivation (*Figure 1B*). By contrast, an SR motif present in $Na_V$1.1 and $Na_V$1.3 limits binding appreciably (*Ahuja et al., 2015*). Thus, we hypothesized that converting either $Na_V$1.2 or $Na_V$1.6 channels from those that contain the YW motif to those that contain an SR motif would alter ASC binding to those channels significantly.

To test for the effects of this motif substitution, we first examined currents generated by wildtype and YW→SR mutated channels expressed in immortalized cell lines (*Figure 1C*). HEK cells were primarily used for most experiments. But due to the known low transfection efficiency of $Na_V$1.6 in HEK cells, a subset of experiments was performed using an ND7/23 cell line engineered to lack most native $Na_V$ conductance (ND7/LoNa$_V$) (*Vanoye et al., 2024*). Both $Na_V$1.2 and $Na_V$1.6 wildtype channels expressed in cell lines were inhibited markedly by 1 µM GNE-4076 (*Figure 1C*). By contrast, SR knock-in mutants continued to flux sodium in the presence of 1 µM GNE-4076. Dose–response curves revealed an $IC_{50}$ of 5.1 nM for wildtype $Na_V$1.2 that increases 365-fold to 1861 nM for mutant SR channels (*Figure 1D*). For $Na_V$1.6 channels, GNE-4076 $IC_{50}$ increased 475-fold from 184 nM to 87 µM with the SR mutant (*Figure 1D*). To assess whether SR mutations affect channel gating properties in the absence of GNE-4076 binding, steady-state activation and inactivation curves were assessed for wildtype or mutant channels (*Figure 1E*). Consistent with prior work studying a similar mutation in $Na_V$1.7 (*Ahuja et al., 2015*), the YW to SR mutation had no effect on slope factor for both voltage-dependent activation or inactivation for either $Na_V$1.2 or $Na_V$1.6 (*Table 2*); however, both voltage-dependent activation and inactivation hyperpolarized by approximately 2 mV for $Na_V$1.6, but not $Na_V$1.2.

While GNE-4076 binds potently to either $Na_V$1.2 (*Figure 1D, F*) or $Na_V$1.7 expressed in HEK cells (*Roecker et al., 2017*), its affinity was lower for $Na_V$1.6 expressed in ND7/LoNa$_V$ cells. To test if this was due to reductions in affinity imposed by the ND7/23 cell line, we examined current in the few HEK cells in which $Na_V$1.6 current could be measured (*Figure 1F*). While peak currents were small (400 pA vs. 3.2 nA for $Na_V$1.2 transfected using identical protocols), we found that 30 nM GNE-4076 exhibited strong block of $Na_V$1.6 (*Figure 1F*), but due to limitations in expression efficiency in HEK cells, we were unable to collect full dose–response curves to determine $IC_{50}$.

Results in heterologous expression systems indicate that YW→SR substitution reduces GNE-4076-binding affinity markedly. Given these results, we constructed transgenic mice containing the YW→SR mutation. Two distinct lines, termed $Scn8a^{SR/SR}$ and $Scn2a^{SR/SR}$, were generated by mutating both alleles of each sodium channel gene (*Figure 1G*). SR/SR mutations were validated by PCR (see Methods), and mice were bred to be homozygous for the mutations. Comparisons were then made between wildtype mice, mice with YW→SR mutations into $Na_V$1.2 or $Na_V$1.6 alone ($2a^{SR/SR}$ or $8a^{SR/SR}$, respectively), or YW→SR mutations in both $Na_V$1.6 and $Na_V$1.2 (dual $Scn8a/2a^{SR/SR}$). In principle, one could determine a concentration of GNE-4076 in which native channels are blocked potently, sparing SR variant channels

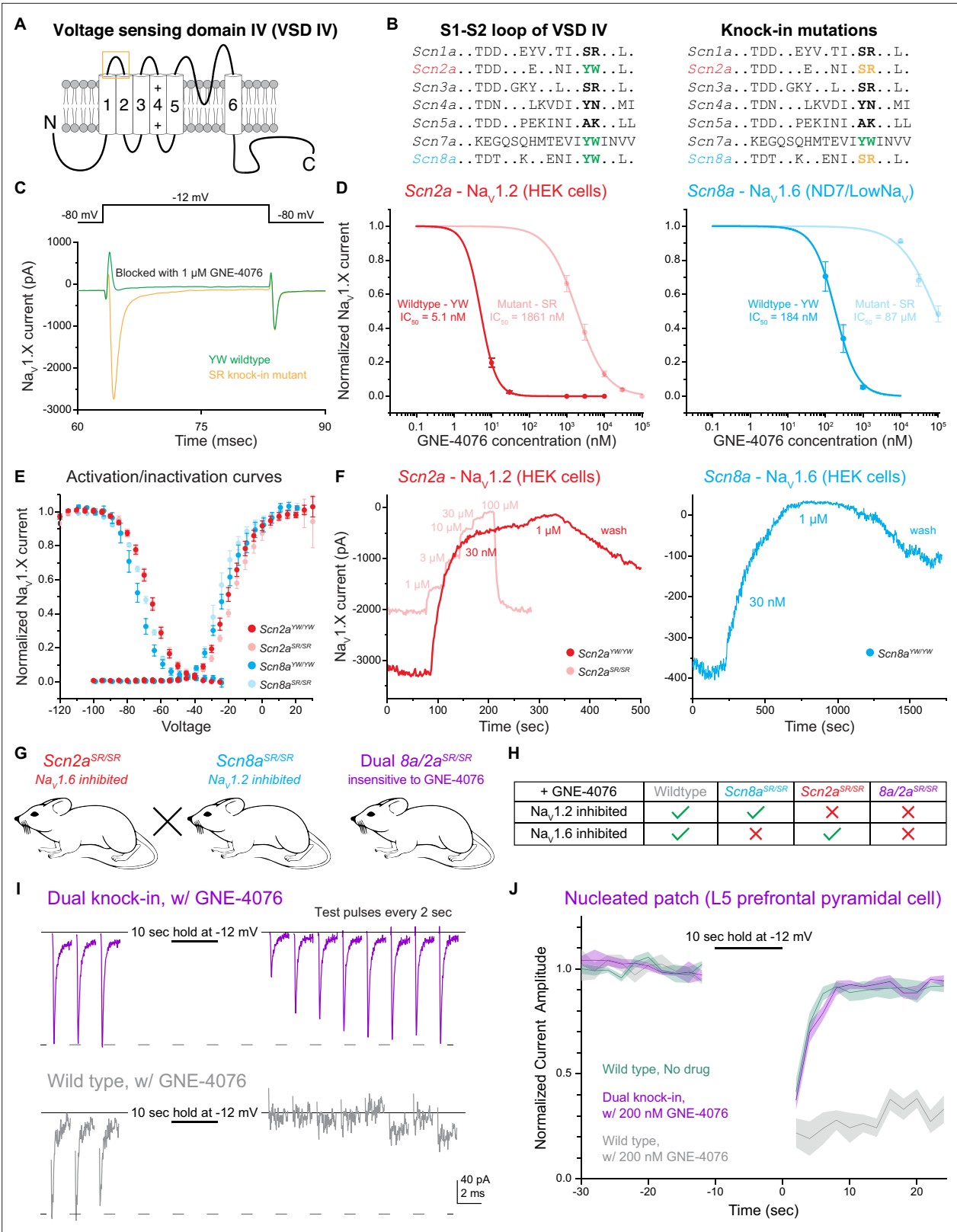

**Figure 1.** The YW motif on Na$_V$1.2 and Na$_V$1.6 increases activity-dependent GNE-4076 potency and subsequent channel inhibition. (**A**) Schematic depicting the fourth voltage sensing domain (VSD-IV) of Na$_V$ isoforms. The six transmembrane spanning regions have high sequence homology among different Na$_V$ isoforms, while linker regions display more sequence divergence. Orange box highlights extracellular S1–S2 loop where aryl sulfonamide compounds (ASCs) are stabilized by a tyrosine–tryptophan (YW) motif. (**B**) Amino acid sequence within the S1–S2 loop of various Na$_V$ isoforms. *Scn2a*

*Figure 1 continued on next page*

*Figure 1 continued*

(Na$_V$1.2) and *Scn8a* (Na$_V$1.6) are the predominant channels expressed in mature, prefrontal pyramidal cells. Both isoforms share a conserved YW sequence that increases ASC potency. Knock-in mutations of *Scn2a* and *Scn8a* were generated by substituting the YW motif with a serine–arginine (SR) sequence present in *Scn1a* and *Scn3a*. (**C**) Example Na$_V$ current traces (pA) of cells expressing either YW wildtype channels or SR knock-in mutant chimeras in the presence of 1 µM GNE-4076. To activate exogenously expressed Na$_V$ channels, cells were held at –80 mV and stepped to –12 mV for 20 ms. GNE-4076 onboarding was performed by holding cells at –12 mV for 10 s. (**D**) Dose–response curves for exogenously expressed *Scn2a* (HEK cells) or *Scn8a* (ND7/LoNa$_V$) in immortalized cell lines. IC$_{50}$ was measured for both YW wildtype channels and SR knock-in mutant chimeras (*Scn2a*$^{YW/YW}$, n = 6; *Scn2a*$^{SR/SR}$, n = 7; *Scn8a*$^{YW/YW}$, n = 8; *Scn8a*$^{SR/SR}$, n = 6). YW→SR knock-in mutations reduced GNE-4076 potency by about 400- to 500-fold relative to wildtype channels. Circles represent normalized mean Na$_V$ current amplitude ± SEM. (**E**) Activation and steady-state inactivation curves for both YW wildtype channels and SR knock-in mutant chimeras. *Scn2a* or *Scn8a* YW→SR mutations alter the efficacy of GNE-4076 while having minor effects on biophysical properties of either isoform. Circles represent mean normalized Na$_V$ current amplitude ± SEM. Unpaired *t*-test with Welch's correction. No significance detected between wildtype and mutant channels for both V1/2 of activation or inactivation. (**F**) Example current amplitude response graphs for Na$_V$1.2 (red) and Na$_V$1.6 (blue) expressed in HEK cells. Cells were perfused with increasing concentrations of GNE-4076 throughout the recording. Individual current response recordings from HEK cells expressing *Scn2a* were robust (3.2 nA), and recordings were reproducible for both YW wildtype channels (red) and SR knock-in mutant chimeras (transparent red). Current responses from cells expressing *Scn8a* were variable, with only a few cells exhibiting channel conductance (400 pA). In select cells expressing *Scn8a*, current amplitude (blue) also decreases substantially with 30 nM GNE-4076 and completely with 1 µM. (**G**) Transgenic mouse lines generated with the YW→SR knock-in mutation present on both *ScnXa* alleles. *Scn2a*$^{SR/SR}$ mice were crossed with *Scn8a*$^{SR/SR}$ mice to generate a dual *Scn8a/2a*$^{SR/SR}$ knock-in mouse. (**H**) Overview of the various transgenic (or wildtype) mouse lines used throughout this study. Application of 200 nM GNE-4076 selectively inhibits Na$_V$ isoforms only containing the YW motif. (**I**) Nucleated patch experiments from prefrontal pyramidal cells performed in wildtype or dual *Scn8a/2a*$^{SR/SR}$ knock-in cells in the presence of 200 nM GNE-4076. Baseline conductance was measured by depolarizing cells from –80 to –12 mV every 2 s for 10 pulses. Na$_V$ channels were inactivated by holding the nucleated patch at –12 mV for 10 s. Test pulses were again acquired during recovery similar to baseline pulses. (**J**) Summary graph of normalized current amplitude from nucleated patch experiments in (**I**). Baseline and recovery test pulses were acquired for at least 20 s before and 25 s after the channel inactivation step. Solid line represents normalized mean Na$_V$ current amplitude ± SEM. Graph also includes wildtype, no drug control nucleated patch experiments (wildtype no drug, n = 4; wildtype + GNE-4076, n = 4; *Scn8a/2a*$^{SR/SR}$ + GNE-4076, n = 4).

The online version of this article includes the following source data and figure supplement(s) for figure 1:

**Source data 1.** Data values for each experiment described in *Figure 1*.

**Figure supplement 1.** Dual knock-in mutants (*Scn8a/2a*$^{SR/SR}$) hyperpolarize action potential (AP) threshold without impairing peak d*V*/d*t*.

**Figure supplement 1—source data 1.** Data values for each experiment described in *Figure 1-figure supplement 1*.

(*Figure 1H*). Thus, these mice should enable selective, acute manipulations of Na$_V$1.2- and Na$_V$1.6-dependent aspects of neuronal excitability using a chemical genetics approach.

Given the disparate binding results in heterologous cell lines, we first tested the effects of a dose that should potently block WT channels but not affect SR variant channels to verify that native channels behave similarly to those expressed in HEK cells. Neocortical pyramidal cell somata are thought to express similar levels of Na$_V$1.2 and Na$_V$1.6 on their membranes (*Spratt et al., 2021*; *Spratt et al., 2019*; *Hu et al., 2009*). Thus, excised patches from this region can provide insight into the block of both isoforms. Acute coronal slices containing mPFC were prepared from WT or *Scn8a/2a*$^{SR/SR}$ mice, and nucleated patches were excised from layer 5 pyramidal cells. After establishing a stable baseline of Na$_V$-mediated current evoked from a holding voltage of –80 mV, a value comparable to the resting membrane potential studied in later current-clamp experiments, cells were pulsed to –12 mV for 10 s to achieve near-complete block of NaV isoforms, then returned to –80 mV (*Figure 1I*). In untreated WT conditions, currents recovered to near-baseline levels within 5 s (*Figure 1J*). A similar quick recovery was observed in cells from *Scn8a/2a*$^{SR/SR}$ mice in the presence of 200 nM GNE-4076. By contrast, WT neurons exposed to the same dosing were inhibited markedly, with no appreciable recovery within 25 s (*Figure 1I, J*). Taken together, these results show that the YW to SR mutation reduces ASC binding and inhibition of Na$_V$ chimeras both in immortalized cell lines and pyramidal neurons.

**Table 2.** Voltage dependence of activation and inactivation for wildtype and mutant Na$_V$ isoforms, related to *Figure 1*.

| Na$_V$ channel | V ½ of activation (mV) | Slope factor | V ½ of inactivation (mV) | Slope factor |
|---|---|---|---|---|
| *Scn2a*$^{YW/YW}$ or wildtype Na$_V$1.2 (n = 6) | –20.1 ± 2.6 | 6.6 ± 0.5 | –68.3 ± 1.8 | 6.9 ± 0.3 |
| *Scn2a*$^{SR/SR}$ or mutant Na$_V$1.2 (n = 7) | –18.7 ± 2.1 | 6.4 ± 0.6 | –68.1 ± 0.8 | 7.1 ± 0.5 |
| *Scn8a*$^{YW/YW}$ or wildtype Na$_V$1.6 (n = 8) | –23.9 ± 2.3 | 6.7 ± 0.4 | –74.6 ± 3.3 | 6.9 ± 0.5 |
| *Scn8a*$^{SR/SR}$ or mutant Na$_V$1.6 (n = 6) | –26.1 ± 3.2 | 6.5 ± 0.4 | –71.9 ± 1.2 | 7.0 ± 0.3 |

**Table 3.** Neuronal action potential (AP) firing properties at baseline, related to *Figures 3 and 4*.

| Na$_V$ channel | Threshold (mV) | Peak dV/dt (V/s) | Spike # | Last AHP |
|---|---|---|---|---|
| No drug control (*n* = 12) | –42.53 ± 0.61 | 555.82 ± 17.57 | 5.83 ± 0.41 | –49.47 ± 0.43 |
| Wildtype (*n* = 11) | –42.41 ± 0.83 | 554.42 ± 14.5 | 5.36 ± 0.36 | –50.04 ± 0.71 |
| *Scn8a/2a$^{SR/SR}$* (*n* = 10) | –44.87 ± 0.47 | 545.11 ± 12.89 | 5.09 ± 0.16 | –51.39 ± 0.5 |
| *Scn2a$^{SR/SR}$* (*n* = 11) | –42.3 ± 0.76 | 520.05 ± 13.51 | 6.09 ± 0.32 | –49.31 ± 0.84 |
| *Scn8a$^{SR/SR}$* (*n* = 11) | –43.87 ± 0.65 | 517.36 ± 21.83 | 5.18 ± 0.55 | –50.46 ± 0.74 |

We also asked whether neuronal AP properties were affected by these mutations in *Scn8a/2a$^{SR/SR}$* mice by assessing both threshold and peak dV/dt in the absence of GNE-4076 (*Figure 1—figure supplement 1*). Compared to wildtype cells, there were no detectable changes to peak dV/dt in *Scn8a/2a$^{SR/SR}$* neurons (*Figure 1—figure supplement 1B*). However, the AP threshold was hyperpolarized in *Scn8a/2a$^{SR/SR}$* neurons, likely due to small changes in the voltage dependence of activation observed for heterologously expressed Na$_V$1.6 (*Figure 1—figure supplement 1A*, *Table 2*). Application with GNE-4076 in either wildtype or *Scn8a/2a$^{SR/SR}$* neurons had no further effect on threshold or peak dV/dt compared to controls recording in the absence of GNE-4076, suggesting minimal drug binding occurs without marked depolarization or spiking activity (*Figure 1—figure supplement 1*, *Table 3*).

## Differential roles for Na$_V$1.6 and Na$_V$1.2 in AP initiation and somatic excitability

Na$_V$1.2 and Na$_V$1.6 are differentially distributed in neuronal arbors and are thought to contribute to different aspects of AP initiation and propagation. In mature neocortical pyramidal cells, Na$_V$1.6 is enriched in the distal AIS, the site where APs initiate (*Spratt et al., 2021*; *Hu et al., 2009*; *Katz et al., 2018*; *Royeck et al., 2008*). Following initiation, APs forward propagate along the axon via Na$_V$1.6-enriched nodes of Ranvier, but also backpropagate into the soma through a region enriched with a mix of Na$_V$1.2 and Na$_V$1.6 (*Hu et al., 2009*). Several studies have sought to identify specific roles for each isoform by conditional deletion of either isoform (*Spratt et al., 2021*; *Katz et al., 2018*). Unfortunately, in such conditions, the residual isoform compensates for loss to some degree, making interpretation of individual isoform roles difficult. We therefore leveraged ASC-based blocks to study acute, differential inhibition of Na$_V$ isoforms to better understand the individual roles of Na$_V$1.2 and Na$_V$1.6 in AP excitability.

Changes to the AP waveform were visualized with phase plane plots, which plot membrane voltage versus membrane voltage velocity (*Figure 2B*, *Figure 3—figure supplement 1A*). Within phase plots, spike threshold appears as a sudden deviation from rest and is defined as the voltage at which voltage velocity (dV/dt) first exceeds 15 V/s (*Figure 3—figure supplement 1A*, right). Following spike threshold, a neuron's voltage transits through two components of Na$_V$ recruitment, first in the AIS and then in the soma (*Baranauskas et al., 2013*; *Kole and Stuart, 2008*). These result in characteristic 'humps' in the depolarizing aspect of the phase plot (*Shu et al., 2007*; *Yu et al., 2008*). Thus, these different components of the phase plot can aid one's understanding of effects on specific components of Na$_V$-mediated excitability.

To provide a priori predictions of potential ASC-based effects on excitability, we constructed a compartmental model in which each channel's density could be modulated. Channels were distributed based on predictions from empirical anatomical and physiological studies (*Spratt et al., 2021*; *Nelson et al., 2024*; *Hu et al., 2009*; *Katz et al., 2018*; *Kole et al., 2008*). Na$_V$1.6 was enriched in the distal AIS, Na$_V$1.2 was enriched in the proximal AIS, both channels were expressed at equal levels in the soma and the first 20 µm of dendrite closest to the soma, and Na$_V$1.2 was expressed exclusively in all other dendrites (*Figure 2A*). Within this model, each channel's density was modulated in 10% increments, from 100 to 0% (*Figure 2B*).

In WT conditions (100% density of both channels), models generated APs with a threshold and AP kinetics comparable to empirical baseline observations (*Figure 2B, C*). Within this model, reducing Na$_V$1.2 or Na$_V$1.6 density had clear, dissociable effects. Progressive reduction of Na$_V$1.6 produced a

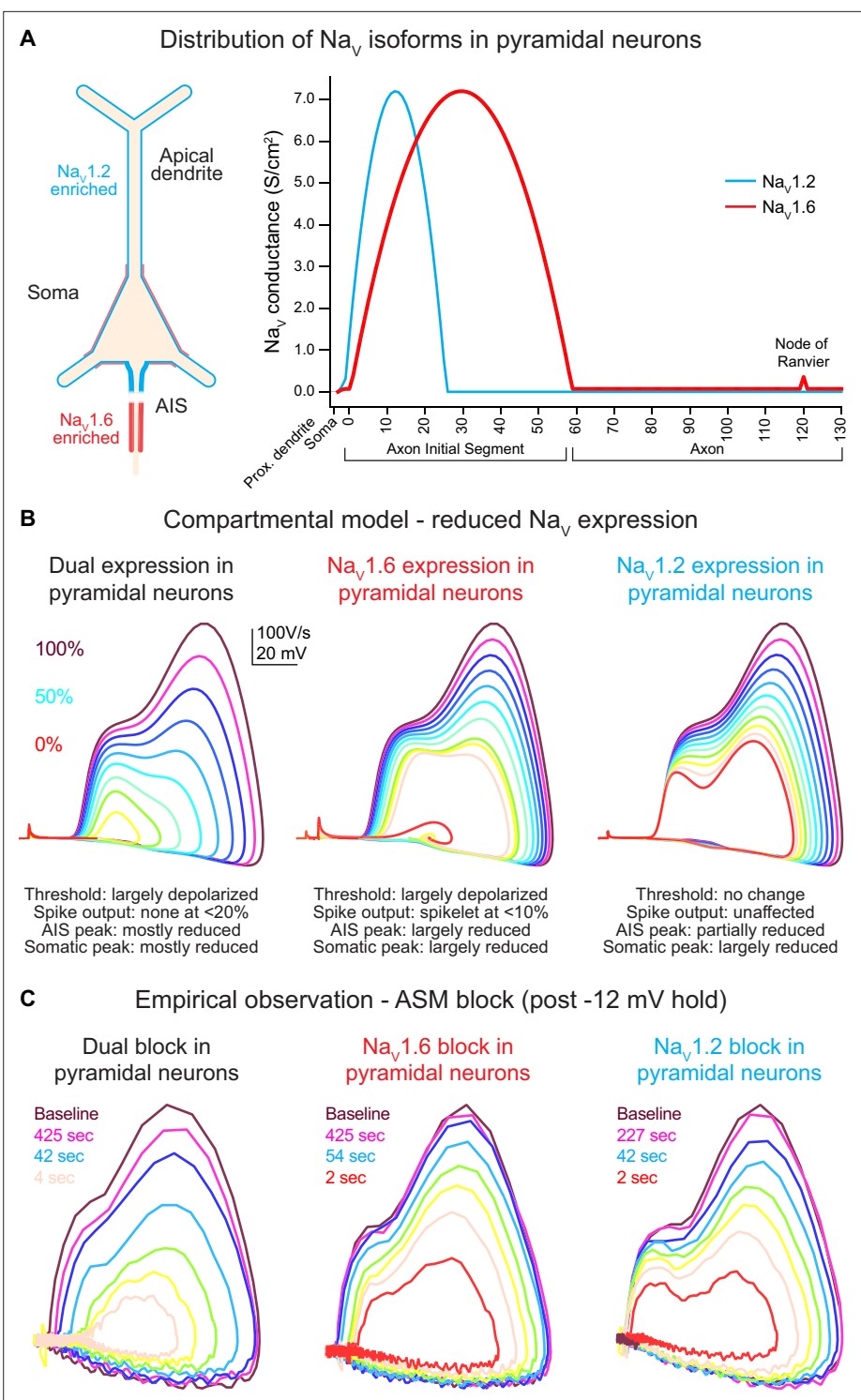

**Figure 2.** Global *Scn2a*, *Scn8a*, or dual loss in compartmental models distinctly impacts key action potential (AP) properties. (**A**) Na$_V$1.6 and Na$_V$1.2 are equally expressed in soma and proximal dendrites. Expression pattern is more distinct in other regions with Na$_V$1.6 enriched in the distal axon initial segment (AIS), axon, and nodes of Ranvier, whereas Na$_V$1.2 is found exclusively in the proximal AIS and distal dendrites. (**B**) Compartmental model representing changes to phase plots when Na$_V$ isoform expression is reduced from 100% (warmer colors) to 0% (in 10% increments) based on known localization across distinct neuronal localities. Lower Na$_V$1.6 expression depolarized spike threshold and decreased both AIS and somatic AP velocity (d$V$/d$t$). Reduced Na$_V$1.2 expression largely impacts backpropagation and somatic AP velocity. (**C**) Empirical observations of phase plot following near-

*Figure 2 continued on next page*

*Figure 2 continued*

complete channel block with ASCs. Darker trace represents phase plot taken at baseline prior to –12 mV hold for 30 s (see *Figure 3A*). Colored traces represent recovery phase plots at different times post –12 mV hold for 30 s with warmer colors depicting more time lapsed and increased channel recovery.

The online version of this article includes the following figure supplement(s) for figure 2:

**Figure supplement 1.** Compartmental model sensitivity analysis.

progressive depolarization of AP threshold and a corresponding decrement in d$V$/d$t$ throughout the entire rising phase of the AP. Na$_V$1.2 reduction, by contrast, had no effect on AP threshold or the initial velocity of the AP initiated in the distal AIS. Instead, components of the AP related to backpropagation and recruitment of somatic Na$_V$ channels were impaired only, with a decrement in peak d$V$/d$t$ and peak AP membrane potential. To validate the accuracy of the parameters used in our model, we also performed sensitivity analysis (*Figure 2—figure supplement 1*). Here, we adjusted the crossover point of Na$_V$1.2 and Na$_V$1.6 within the AIS as well as varied the density ratio of both channels (*Figure 2—figure supplement 1A, B*). Increasing the Na$_V$1.6 ratio to 100% consistently hyperpolarized AP threshold across all AIS crossover positions (*Figure 2—figure supplement 1B, C*). However, d$V$/d$t$ was highly variable and random when either the AIS crossover point is shifted or Na$_V$ ratios are altered.

While these models provide clues to the differential roles of Na$_V$1.2 and Na$_V$1.6, we needed to design an experimental strategy that increased the overall degree of Na$_V$ blockade using ASCs in mice where either Na$_V$1.6 or Na$_V$1.2 was mutated to be insensitive to 200 nM GNE-4076 binding (*Figures 1G and 3A*). Since it is nearly impossible to achieve full block with ASCs under physiological firing, we developed a hybrid current- and voltage-clamp experiment to study the effects of more complete isoform blockade (*Figure 3B*). In this protocol, neurons were held to –80 mV in current clamp with constant bias current (if necessary), and baseline APs were elicited with brief somatic current injection (300 ms, amplitude adjusted to evoke ~4–5 APs). We then promoted Na$_V$ activation and inactivation by voltage-clamping neurons to –12 mV for 30 s (*Figure 3A*). Following voltage-clamp, neurons were returned to current-clamp with the same bias current as used in baseline conditions. We then evoked APs, first at an interstimulus interval of 2 s, then at successively increasing intervals of 5–60 s, allowing for somewhat consistent sampling on a log-base timescale that aligns well with both channel recovery from inactivation and recovery from GNE-4076 block (*Figure 3D, F*). For clarity and simplicity, we will describe studies based on which channel is inhibited rather than which channel was rendered insensitive to GNE-4076. For example, a study in the *Scn2a$^{SR/SR}$/Scn8a$^{+/+}$* animal is a case where Na$_V$1.6 can be blocked.

Empirical data at various time points post –12 mV hold reveals that recovery from ASC-dependent block increases channel availability from approximately 0–10% to about 90–95% of the baseline spike when more time has elapsed (*Figures 2C and 3B, C*). This recovery in available channels mimics changes observed to phase plots in compartmental models based on overall channel expression (*Figure 2B*).

We then focused our analysis on the recovery of threshed or peak d$V$/d$t$ across all genotypes (*Figure 3*). In untreated WT cells, AP threshold following voltage-clamp recovered to baseline values immediately (*Figure 3C, D*). By contrast, peak somatic d$V$/d$t$ recovered more slowly (*Figure 3C, F*), likely reflecting recovery from slow inactivation of channels in the perisomatic region (*Colbert et al., 1997*; *Park et al., 2013*; *Toib et al., 1998*). Peak d$V$/d$t$ recovered to within 87.12 ± 0.86% by 12–20 s of voltage-clamp offset (*Figure 3F, G*). Subsequent recovery of the residual peak d$V$/d$t$ was slower, taking another 15–30 s to recover the next 5% of peak d$V$/d$t$ (*Figure 3F*). Identical results were observed in *Scn8a/2a$^{SR/SR}$* cells treated with 200 nM GNE-4076, suggesting that any effect observed in single knock-in recordings will be due to block of GNE-4076 sensitive channels (*Figure 3B–G*). We therefore focused on the 12–20 s after voltage-clamp offset for subsequent analysis, as it is a period in which most channel-intrinsic recovery has occurred, but also a period in which we would still expect significant block from GNE-4076.

When individual channel isoforms were blocked more completely with voltage steps to –12 mV, dramatic changes in AP threshold and peak d$V$/d$t$ were observed. When Na$_V$1.6 was blocked, threshold depolarized by 5.8 ± 0.7 mV (*Figure 3E*). When Na$_V$1.2 was blocked instead, threshold was unaffected (Δ $V_m$: –0.1 ± 0.2). Thus, AP threshold and AP initiation appear to be initiated in an Na$_V$1.6-rich region

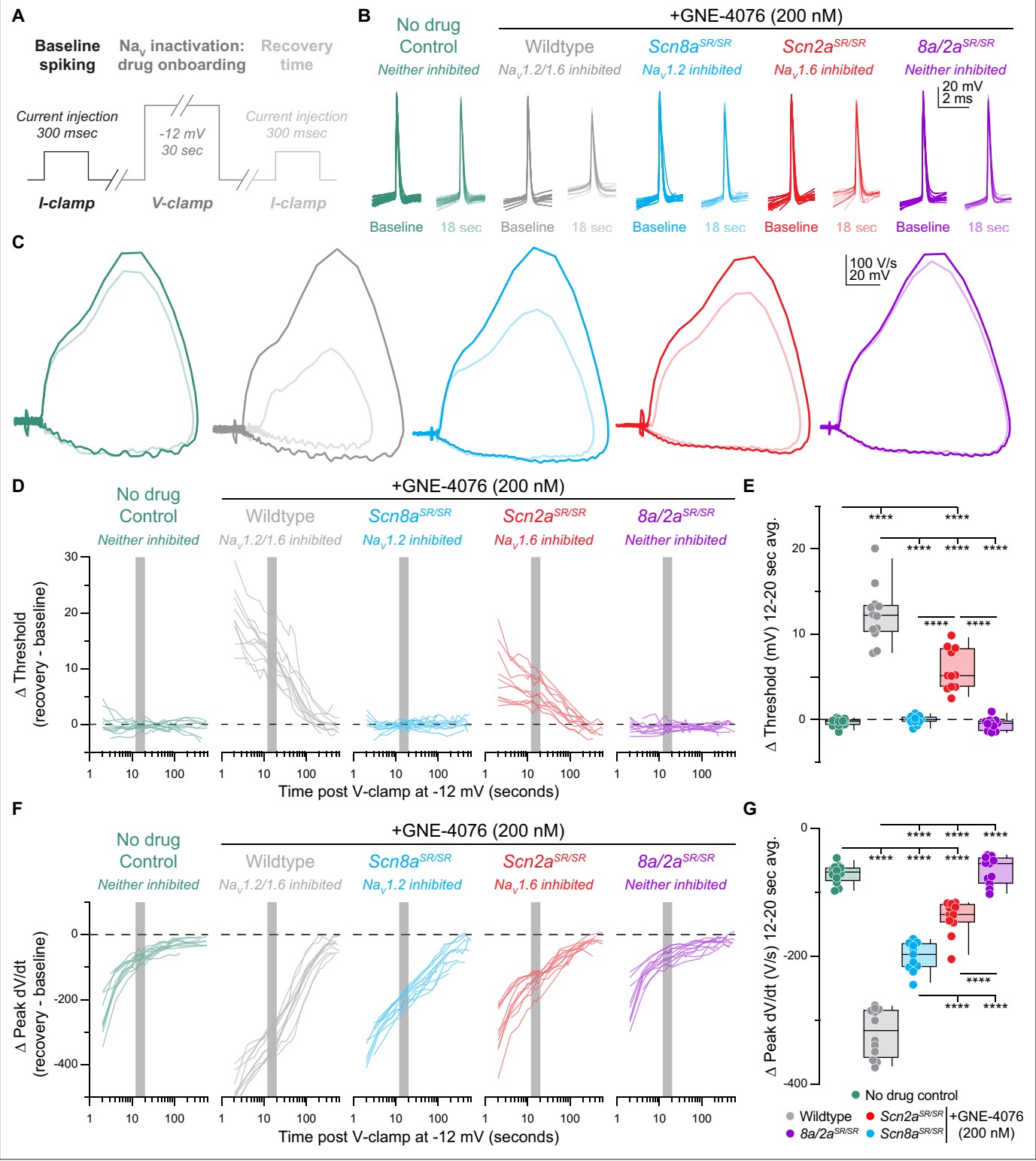

**Figure 3.** Recovery of action potential (AP) firing properties is greatly diminished following dual Na$_V$1.6 and Na$_V$1.2 inhibition compared to selective block of individual channels. (**A**) Protocol used to characterize recovery of AP firing properties. Baseline spiking is determined by injecting current for 300 ms to elicit five to six APs. To promote Na$_V$ inactivation and maximal GNE-4076 onboarding, neurons are held at –12 mV in voltage-clamp for 30 s. Recovery of AP firing is evaluated by injecting the same current stimulus defined during baseline spiking with an inter-stimulus interval starting at 2 s, followed by 5, 15, 30, and 60 s. (**B**) Overlaid waveform of first AP at baseline or 18 s post GNE-4076 onboarding for all conditions (wildtype no drug, *n* = 12; wildtype + GNE-4076, *n* = 11; *Scn8a/2a^SR/SR^* + GNE-4076, *n* = 10; *Scn2a^SR/SR^* + GNE-4076, *n* = 11; *Scn8a^SR/SR.^* + GNE-4076, *n* = 11). (**C**) Overlaid phase

*Figure 3 continued on next page*

*Figure 3 continued*

plane of AP traces at baseline (100% transparency) or 18 s post GNE-4076 onboarding (20% transparency) for each condition in (**B**). Plots represent the AP velocity by taking the first derivative (d*V*/d*t*, *y*-axis) versus the membrane potential (mV, *x*-axis). Colors are matched to conditions represented in (**B**). (**D**) Recovery of AP threshold ($V_m$) represented as a delta value for individual cells plotted against time post GNE-4076 onboarding (log-scale). For Δ $V_m$, the baseline value is subtracted from individual timepoints throughout the recovery phase (Δ mV = recovery timepoint – baseline). Colors are matched to conditions represented in (**B**). Gray shaded bar represents recovery between 12 and 20 s. (**E**) Summary data for Δ $V_m$ at 12–20 s post GNE-4076 onboarding (time period represented as gray bar in (**D**)). Box plots show median and 90% tails. Circles represent individual cells. One-way ANOVA, Holm–Šídák multiple comparisons test. ****p < 0.0001. (**F**) Recovery of AP peak velocity (d*V*/d*t*) represented as a delta value for individual cells plotted against time post GNE-4076 onboarding (log-scale). For Δ d*V*/d*t*, the baseline value is subtracted from individual time points throughout the recovery phase (Δ V/s = recovery timepoint – baseline). Colors are matched to conditions represented in (**B**). Gray shaded bar represents recovery between 12 and 20 s. (**G**) Summary data for Δ d*V*/d*t* at 12–20 s post GNE-4076 onboarding (time period represented as gray bar in (**F**)). Box plots show median and 90% tails. Circles represent individual cells. One-way ANOVA, Holm–Šídák multiple comparisons test. ****p < 0.0001.

The online version of this article includes the following source data and figure supplement(s) for figure 3:

**Source data 1.** Data values for each experiment described in *Figure 3*.

**Figure supplement 1.** Depolarization across the axon initial segment (AIS) compartment is mediated by distinct Na$_V$ isoforms.

**Figure supplement 1—source data 1.** Data values for each experiment described in *Figure 3- figure supplement 1*.

**Figure supplement 2.** Recovery of action potential (AP) firing properties represented as absolute values.

**Figure supplement 2—source data 1.** Data values for each experiment described in *Figure 3-figure supplement 2*.

---

in control conditions; but when Na$_V$1.6 is inhibited, APs can occur at more depolarized potentials, likely mediated predominately by Na$_V$1.2.

To examine these relative contributions further, we analyzed the rising phase of APs more closely (***Figure 3—figure supplement 1A***). As described above, the first component of the rising phase of the AP reflects recruitment of Na$_V$ channels localized to the AIS (***Figure 3—figure supplement 1A***, middle). Based on changes in voltage acceleration (second derivative) within this period, the AIS component can be further divided into initiation and AIS backpropagation components (***Hu et al., 2009***; ***Favero et al., 2018***; ***Figure 3—figure supplement 1A***, right). These periods were divided based on a time point during the AIS phase where voltage acceleration first peaked (AIS inflection point) or afterwards decreased, creating a trough in an acceleration versus time graph (AIS max; ***Figure 3—figure supplement 1B***). The d*V*/d*t* at the AIS inflection point was affected only by Na$_V$1.6 block (***Figure 3—figure supplement 1C***), whereas the d*V*/d*t* at the trough was affected by block of either isoform (***Figure 3—figure supplement 1D, E***). This reinforces the model where APs are initiated via the Na$_V$1.6-enriched distal AIS, and where the depolarization produced by these distally localized channels recruits Na$_V$1.2 in the more proximal AIS. Absolute values are also reported for threshold, peak d*V*/d*t*, and AIS max for all recorded cells in recovery experiments (***Figure 3—figure supplement 2A–F***).

## Acute block of Na$_V$1.2 increases pyramidal cell AP output

Previously, we showed that conditional knockout of *Scn2a* increased AP output (***Spratt et al., 2021***). We hypothesized that this was due to the lack of Na$_V$1.2 in dendrites. When they are absent from this compartment, the dendrite does not depolarize as effectively, leading to a corresponding decrease in the recruitment of dendrite-localized potassium channels (K$_V$). Consequently, neurons repolarize less between APs, making it easier to evoke the next AP in the Na$_V$1.6-enriched AIS.

Though this hypothesis was supported by compartmental modeling demonstrating that acute block of Na$_V$1.2 mirrored empirical observations, we could not eliminate the possibility that some form of cellular compensation occurred in the weeks between conditional knockout induction and acute slice experiments (***Miralles and Patel, 2022***). Therefore, we leveraged GNE-4076 to test the effects of acute Na$_V$1.2 block (***Figure 4A***), eliminating the possibility of genetic or post-translational compensation. We further compared this manipulation to block of Na$_V$1.6 or both isoforms by examining overall AP output from protocols described in ***Figure 3A***.

Remarkably, acute Na$_V$1.2 block mirrored conditional knockout and was the only condition in which AP output increased (***Figure 4B***) and was associated with a depolarization in afterhyperpolarization voltage (***Figure 4C***). In contrast, Na$_V$1.6 block decreased AP output, as did block of both channels (***Figure 4B***). This indicates that blocking Na$_V$1.2 alone can increase AP output, independent of compensatory changes to other channels that may occur with genetic manipulations.

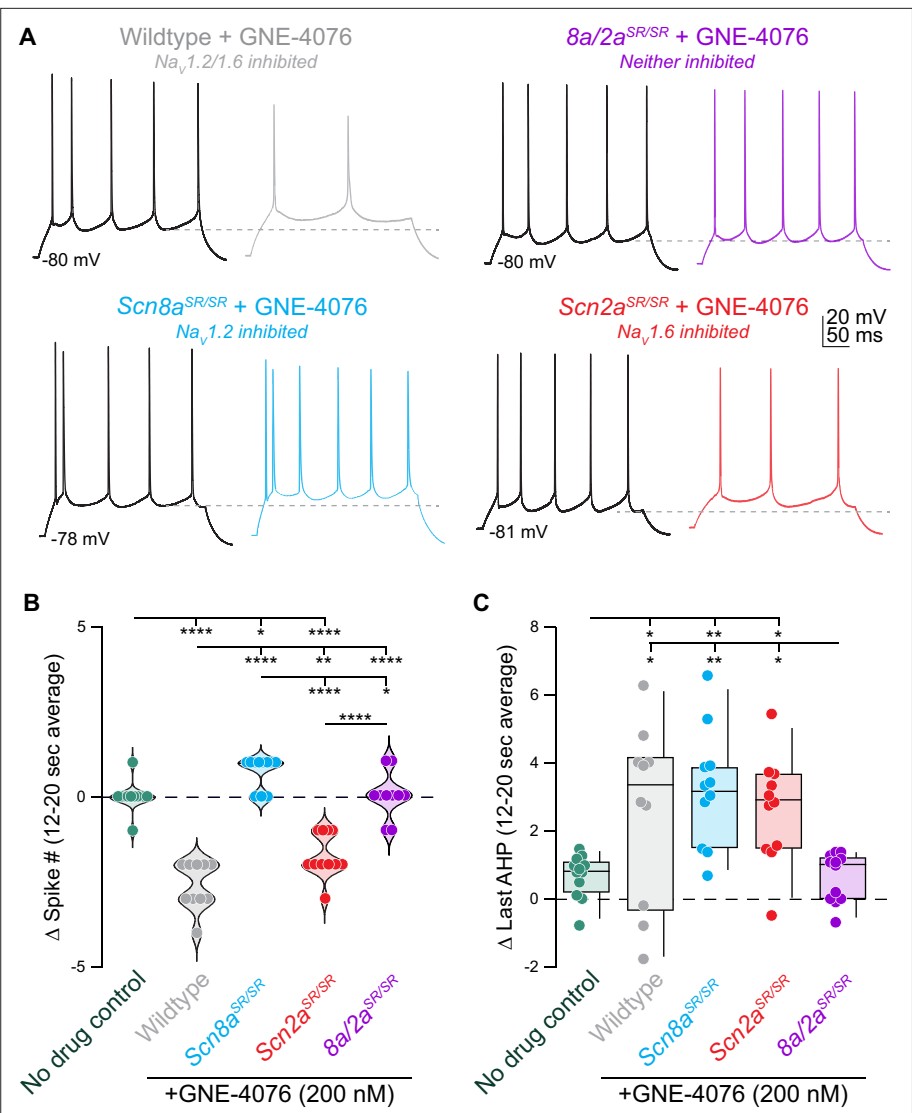

**Figure 4.** Acute inhibition of Na$_V$1.2 increases action potential (AP) excitability. (**A**) AP train over 300 ms at baseline (black) or 18 s post GNE-4076 onboarding (color) for all conditions (wildtype no drug, n = 12; wildtype + GNE-4076, n = 10; *Scn8a/2a$^{SR/SR}$* + GNE-4076, n = 11; *Scn2a$^{SR/SR}$* + GNE-4076, n = 11; *Scn8a$^{SR/SR}$* + GNE-4076, n = 11). Dashed line represents $V_m$ of last afterhyperpolarization (AHP). (**B**) Summary data for Δ spike number at 12–20 s post GNE-4076 onboarding. Violin plots show median. Circles represent individual cells. One-way ANOVA, Holm–Šídák multiple comparisons test. *p < 0.05, **p < 0.01, ****p < 0.0001. (**C**) Summary data for Δ last AHP at 12–20 s post GNE-4076 onboarding. Box plots show median and 90% tails. Circles represent individual cells. One-way ANOVA, Holm–Šídák multiple comparisons test. *p < 0.05, **p < 0.01.

The online version of this article includes the following source data for figure 4:

**Source data 1.** Data values for each experiment described in *Figure 4*.

## Activity-dependent effects of ASCs on Na$_V$ channels and subsequent inhibition alter AP properties and firing rate

During physiological activity, non-selective pharmacological inhibition of Na$_V$ channels in neurons greatly reduces cellular excitability and AP firing (*Tukker et al., 2023*; *Thouta et al., 2022*). ASCs are unique in that they require prolonged bursts of activity or even neuronal hyperexcitability to stabilize more channels in the inactivated state (*Ahuja et al., 2015*; *Goodchild et al., 2024*; *Johnson et al., 2024*; *Johnson et al., 2022*). In contrast to the data presented above, where we observe near-complete block immediately following prolonged depolarization (*Figure 3—figure*

*supplement 2C*, absolute values), neurons experiencing physiological levels of activity likely never reach high percentages of channel blockade. Thus, in normal spiking neurons, ASCs would therefore be predicted to exhibit use dependence, progressively blocking channels in proportion to a neuron's activity rate.

To test this concept, we generated prolonged periods of activity in current-clamp by injecting 300 pA into the somatic pipette for 10 s (*Figure 5A and B*). Drug-naive WT neurons responded to somatic current injection with repetitive spiking at a rate of 14.7 ± 0.5 Hz. Neurons fired at a steady state after the first second, with stable AP threshold and instantaneous AP frequency (*Figure 5C*). Spiking characteristics were identical in *Scn8a/2a^{SR/SR}* cells treated with GNE-4076, indicating that, at 200 nM, neurons are insensitive to GNE-4076 when the YW→SR motif substitution is present (*Figure 5C, D*). However, in WT neurons treated with GNE-4076, threshold continued to depolarize after the first second, with corresponding decrements in instantaneous frequency and peak AP d$V$/d$t$ throughout the stimulus (*Figure 5C, D*). This demonstrates that GNE-4076 suppresses firing of neurons with a baseline firing rate of ~15 Hz (*Table 4*).

We then repeated experiments described above for single mutated channels—where APs were generated with somatic depolarization over 10 s (*Figure 6A, B*). In cases where Na$_V$1.6 alone could be inhibited, AP threshold depolarized in response to GNE-4076 as much as in WT cases (*Figure 6C, D*). Furthermore, AP frequency and peak somatic AP d$V$/d$t$ were reduced to levels observed in WT cells (*Figure 6C, D*). In cases where Na$_V$1.2 alone could be inhibited, changes in AP threshold and instantaneous frequency were no different than *Scn8a/2a^{SR/SR}* conditions. Indeed, the only change in AP properties was a decrease in peak AP d$V$/d$t$ (*Figure 6D*). This suggests that AP threshold and instantaneous frequency are more sensitive to Na$_V$1.6 antagonism, whereas peak somatic d$V$/d$t$ can be affected by antagonism of either isoform. Absolute values are also reported for threshold, peak d$V$/d$t$, and instantaneous frequency for all recorded cells in prolonged activity burst experiments (*Figure 6—figure supplement 1A*).

## Leveraging use-dependent, isoform-selective Na$_V$ pharmacology as anticonvulsants

Epilepsy is often associated with aberrant, excessive excitability in neocortical networks, whether from direct hyperexcitability in pyramidal cells (*Lopez-Santiago et al., 2017*; *Sanders et al., 2018*) or disinhibition of pyramidal cells via alterations in inhibitory networks (*Favero et al., 2018*; *Tran et al., 2020*). Several use-dependent Na$_V$ antagonists with structures similar to GNE-4076 are being developed, exhibiting differential selectivity for Na$_V$1.2 and Na$_V$1.6 (*Goodchild et al., 2024*). Use dependence may have advantages as anti-epileptics. In theory, they would have minimal effect on neurotypical network activity unless such networks were hyperactive enough to promote drug binding. This may occur preferentially in seizure states. But given the results above demonstrating that block of Na$_V$1.2 and Na$_V$1.6 can have different effects on overall AP output, it is critical to determine how block of either channel affects overall activity in seizure-like conditions.

To test this, we mimicked synaptic input by repeatedly injecting a 60-s-long somatic current composed of Poisson-distributed EPSC and IPSC-like waveforms designed to elicit ~10 Hz spiking in baseline conditions (*Figure 7A*). Following this baseline, the PSC-like train was repeated, imposed upon a 400-pA standing current injection to increase spiking to ~30 Hz (*Figure 7C*), mimicking spike rates commonly observed during cortical seizure (*Necklemann et al., 1998*; *Timofeev and Steriade, 2004*). After this seizure-like event, cells were returned to baseline voltages, and recovery was assessed with four more repetitions of the 60 s PSC stimulus (*Figure 7A–C*).

In the presence of 200 nM GNE-4076, baseline firing rates were mostly stable across all conditions (*Figure 7—figure supplement 1A*). Wildtype, no drug and *Scn8a/2a^{SR/SR}* conditions observed no change in firing rate, while dual or selective Na$_V$1.6 blockade saw a slight decrease by 60 s of postsynaptic activity (*Figure 7—figure supplement 1B*). Interestingly, we did see a slight increase in baseline firing by 60 s when Na$_V$1.2 was selectively targeted (*Figure 7—figure supplement 1A, B*), further highlighting a paradoxical shift to hyperactive neurons when Na$_V$1.2 availability decreases. By contrast, a slight depolarization in AP threshold was observed at baseline when both channels were sensitive to GNE-4076 (e.g., WT plus drug) and to a lesser extent with selective block of Na$_V$1.6 (*Figure 7—figure supplement 1A, B*). Given that this dose can completely block both Na$_V$1.2 and Na$_V$1.6 in HEK cells, a small amount of drug block was expected at baseline.

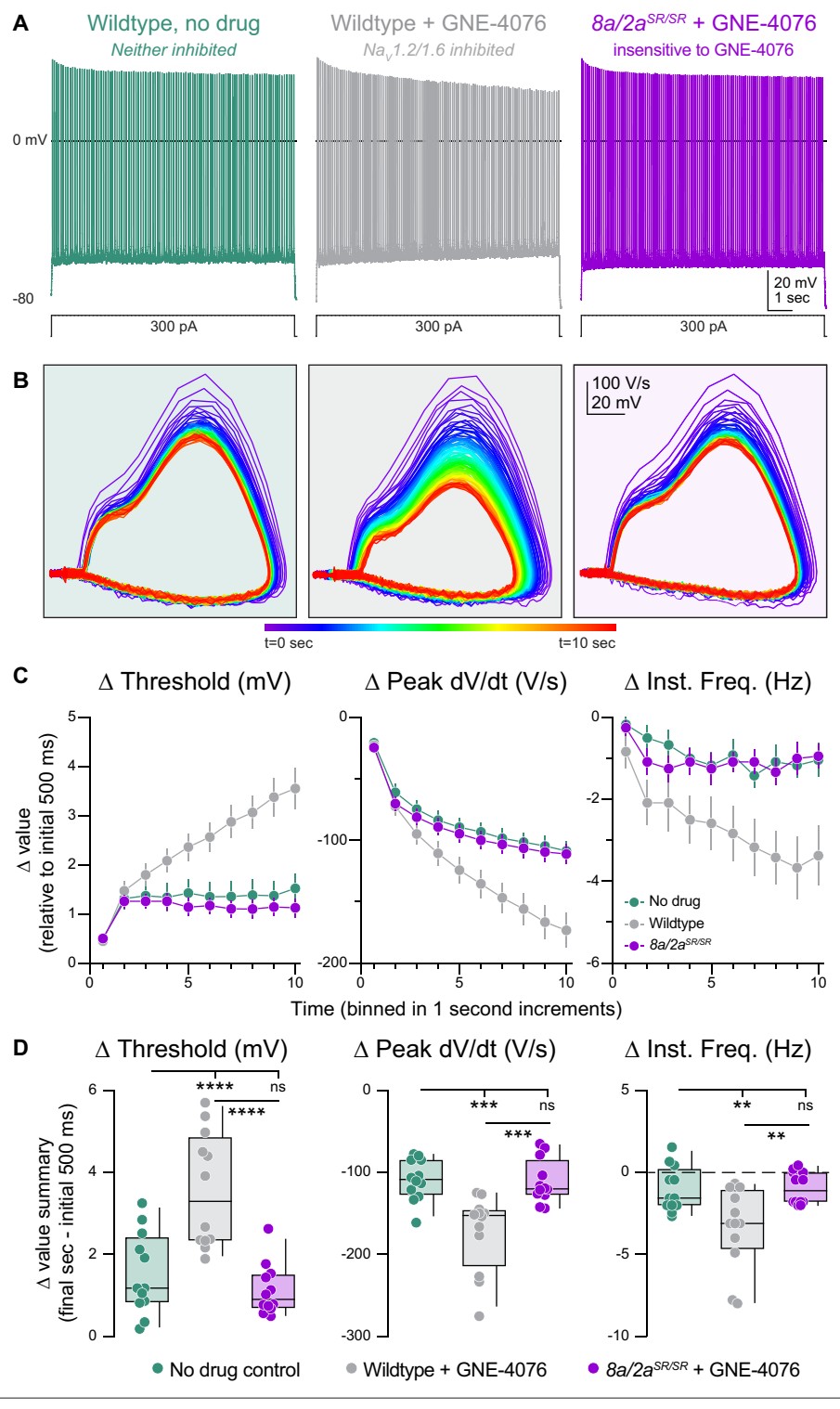

**Figure 5.** Activity-dependent onboarding of GNE-4076 to Na$_V$ channels alters action potential (AP) firing properties in layer 5b, thick-tufted excitatory neurons. (**A**) Representative AP firing response to 300 pA current injection for 10 s in wildtype or *Scn8a/2a$^{SR/SR}$* cells with or without 200 nM GNE-4076. (**B**) Phase plane of AP traces shown in (**A**). Plots represent AP velocity by taking the first derivative (d*V*/d*t*, *y*-axis) versus the membrane potential (mV, *x*-axis). To represent changes with phase plane relative to time, a rainbow color spectrum is used with warmer colors representing more time lapsed. (**C**) Delta threshold (Δ mV), delta peak d*V*/d*t* (Δ V/s), and delta instantaneous firing frequency (Δ Hz) binned in 1-s increments normalized to the initial 500 ms of current injection (binned time

*Figure 5 continued on next page*

*Figure 5 continued*

– initial 500 ms). Circles represent mean Δ value ± SEM. Two-way ANOVA, Holm–Šídák multiple comparisons test. (**D**) Summary data for the final sec in (**C**). Delta values are normalized to the initial 500 ms of the stimulus (binned time – initial 500 ms). Box plots show median and 90% tails. Circles represent individual cells (wildtype no drug, *n* = 12; wildtype + GNE-4076, *n* = 12; *Scn8a/2a*$^{SR/SR}$ + GNE-4076, *n* = 12). One-way ANOVA, Holm–Šídák multiple comparisons test. **p < 0.01, ***p < 0.001, ****p < 0.0001.

The online version of this article includes the following source data for figure 5:

**Source data 1.** Data values for each experiment described in *Figure 5*.

Seizure-like activity induction depolarized AP threshold markedly in all genotypes, and threshold depolarization persisted into the recovery phase for all neurons experiencing Na$_V$1.6 or dual inhibition (*Figure 7C, D*). Seizure-like activity increased spike rate across all genotypes at the onset of seizure-like activity induction (*Figure 7C, E*). All cells exhibited some degree of AP accommodation during this stimulus, including drug-naive controls and cases where both Na$_V$1.2 and Na$_V$1.6 were insensitive to GNE-4076 block. WT cells exposed to GNE-4076 accommodated markedly, returning to baseline firing rates at the end of the seizure-like stimulus (*Figure 7C, E*). A less dramatic effect was noted when Na$_V$1.6 was blocked alone (*Figure 7E*). By contrast, cells where Na$_V$1.2 could be blocked alone were not appreciably different than untreated controls (*Figure 7E*). Furthermore, some cells that remained hyperactive with selective Na$_V$1.2 block compared to control cells throughout the recovery phase. Together, these data suggest that use-dependent pharmacology that targets Na$_V$1.2 and Na$_V$1.6 may be most beneficial if designed for higher potency at Na$_V$1.6.

## Discussion

Here, we combined genetic and pharmacological strategies to transiently, selectively, and reversibly inhibit either Na$_V$1.6 or Na$_V$1.2 function using activity-dependent ASCs, which bind channels in the inactivated state. We show that acute blockade of either isoform has opposing effects on neuronal output: inhibition of Na$_V$1.6 decreases AP output, whereas Na$_V$1.2 increases AP output. Given this, we found that the block of Na$_V$1.6 rather than Na$_V$1.2 was more effective at tempering spiking activity in cells driven to seizure-like levels of AP output. This suggests that pharmacology tuned to preferentially block Na$_V$1.6 over Na$_V$1.2 in a use-dependent manner may be useful as an anti-epileptic.

### Functional implications of differential compartmental Na$_V$ isoform expression

Electrophysiological recordings of AP propagation delays between somatic and axonal recordings demonstrate that APs initiate ~35–50 μm from the soma, in the distal AIS (*Shu et al., 2007*; *Palmer and Stuart, 2006*; *Baranauskas et al., 2013*; *Hu et al., 2009*). This region is enriched with Na$_V$1.6 (*Hu et al., 2009*; *Tian et al., 2014*; *Royeck et al., 2008*; *Lorincz and Nusser, 2008*; *Akin et al., 2015*). Consistent with this, we find that inhibition of Na$_V$1.6 alone alters AP initiation, with an increase in AP threshold and a decrease in total AP number from block of this isoform. Following distal AIS AP initiation, APs sequentially propagate across different neuronal compartments to influence local excitability (*Shu et al., 2007*; *Palmer and Stuart, 2006*; *Baranauskas et al., 2013*). Forward propagation was not examined here, but prior reports demonstrate that it is supported almost exclusively by Na$_V$1.6

**Table 4.** Action potential (AP) firing properties of first spike in 10 s AP train, related to *Figures 5 and 6*.

| Na$_V$ channel | Threshold (mV) | Peak dV/dt (V/s) |
|---|---|---|
| Wildtype, no drug (*n* = 12) | –43.1 ± 0.4 | 549.1 ± 8.8 |
| Wildtype + GNE-4076 (*n* = 12) | –41.7 ± 0.7 | 541.0 ± 12.5 |
| *Scn8a/2a*$^{SR/SR}$ + GNE-4076 (*n* = 12) | –43.7 ± 0.4 | 561.3 ± 14.5 |
| *Scn2a*$^{SR/SR}$ + GNE-4076 (*n* = 12) | –40.7 ± 0.4 | 500.7 ± 12.5 |
| *Scn8a*$^{SR/SR}$ + GNE-4076 (*n* = 11) | –43.3 ± 0.5 | 542.9 ± 11.6 |

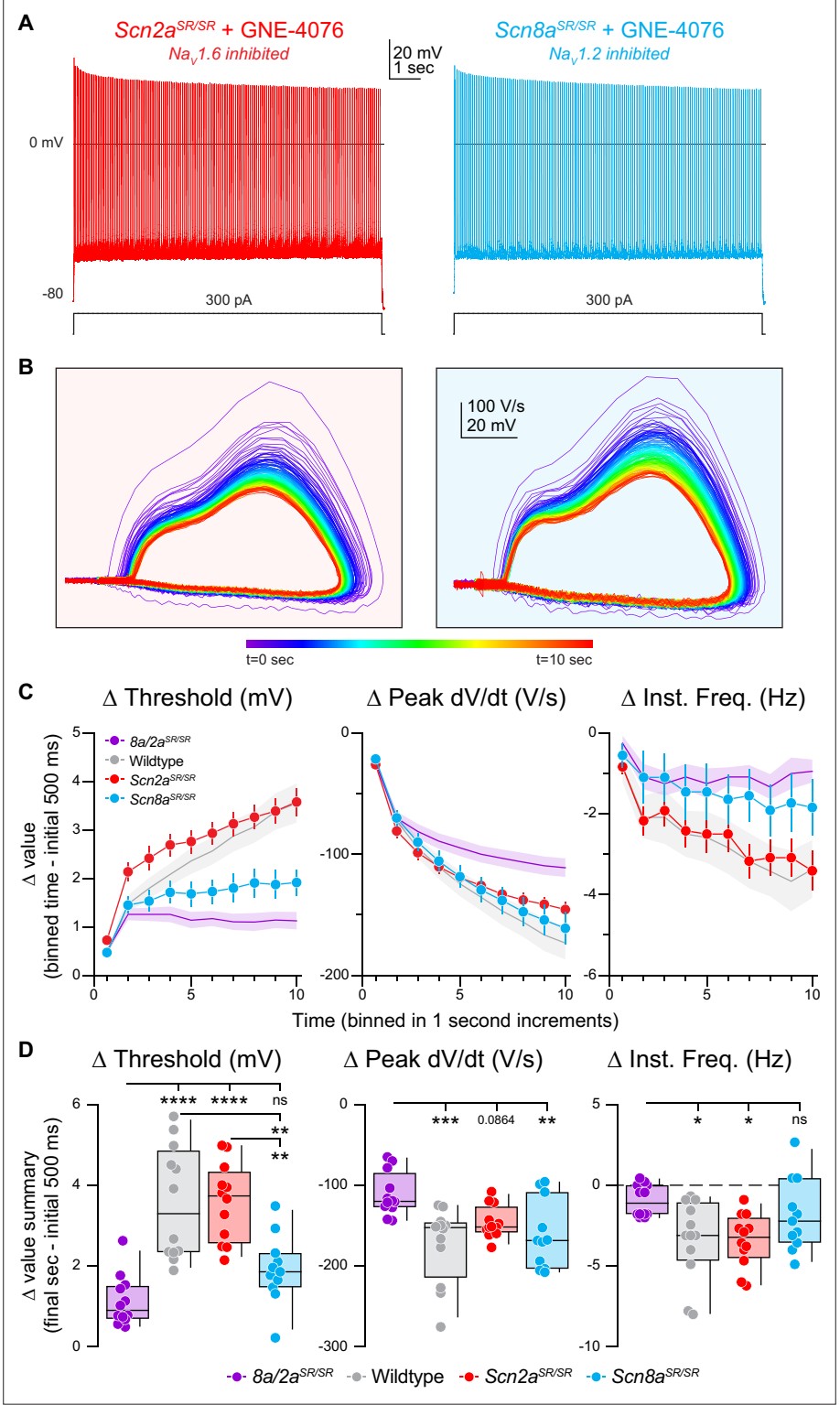

**Figure 6.** Selective inhibition of Na$_V$1.6 depolarizes action potential (AP) threshold markedly, while blocking both Na$_V$1.6 and Na$_V$1.2 reduces AP velocity. (**A**) Representative AP firing response to 300 pA current injection for 10 s in *Scn2a*$^{SR/SR}$ or *Scn8a*$^{SR/SR}$ cells with 200 nM GNE-4076 to selectively inhibit Na$_V$1.6 or NaV1.2, respectively. (**B**) Phase plane of AP traces shown in (**A**). Plots represent AP velocity by taking the first derivative (d*V*/d*t*, *y*-axis) versus the membrane potential (mV, *x*-axis). To represent changes with phase plane relative to time, a rainbow color spectrum is used with warmer colors representing more time lapsed. (**C**) Delta threshold (Δ mV), delta peak d*V*/d*t*

*Figure 6 continued on next page*

*Figure 6 continued*

(Δ V/s), and delta instantaneous firing frequency (Δ Hz) binned in 1-s increments normalized to the initial 500ms of current injection (binned time – initial 500 ms). Circles represent mean Δ value ± SEM. Average Δ value ± SEM for *Scn8a/2a$^{SR/SR}$* + GNE-4076 and wildtype + GNE-4076 from **Figure 5C** are represented. (**D**) Summary data for the final sec in (**C**). Delta values are normalized to the initial 500 ms of the stimulus (binned time – initial 500 ms). Box plots show median and 90% tails. Circles represent individual cells (*Scn8a/2a$^{SR/SR}$* + GNE-4076, $n = 12$; wildtype + GNE-4076, $n = 12$; *Scn2a$^{SR/SR}$* + GNE-4076, $n = 12$; *Scn8a$^{SR/SR}$* + GNE-4076, $n = 11$). One-way ANOVA, Holm–Šídák multiple comparisons test. *$p < 0.05$, **$p < 0.01$, ***$p < 0.001$, ****$p < 0.0001$.

The online version of this article includes the following source data and figure supplement(s) for figure 6:

**Source data 1.** Data values for each experiment described in *Figure 6*.

**Figure supplement 1—source data 1.** Data values for each experiment described in *Figure 6-figure supplement 1*.

**Figure supplement 1.** Absolute values for activity-dependent onboarding of GNE-4076.

(**A**) Absolute values for threshold (mV), peak d$V$/d$t$ (V/s), and instantaneous firing frequency (Hz) binned in 1-s increments. Circles represent mean absolute value ± SEM (wildtype no drug, $n = 12$; wildtype + GNE-4076, $n = 12$; Scn8a/2a$^{SR/SR}$ + GNE-4076, $n = 12$; Scn2a$^{SR/SR}$ + GNE-4076, $n = 12$; Scn8a$^{SR/SR}$ + GNE-4076, $n = 11$). Two-way ANOVA, Holm–Šídák multiple comparisons test.

in nodes of Ranvier and boutons (**Tian et al., 2014**; **Caldwell et al., 2000**; **Kaplan et al., 2001**). By contrast, backpropagation through the proximal AIS and soma recruits a mix of Na$_V$1.6 and Na$_V$1.2.

Prior work has suggested that Na$_V$1.2 channels localized to the proximal AIS are critical for backpropagation, with modeling suggesting that APs would fail to effectively backpropagate into the soma in the absence of proximal AIS Na$_V$1.2 channels (**Hu et al., 2009**). While some efforts have been made to test this empirically with conditional *Scn2a* knockout, such experiments are imperfect, as Na$_V$1.6 redistributes in the AIS of cells that lack Na$_V$1.2 (**Spratt et al., 2021**). Here, we were able to test the role of Na$_V$1.2 in backpropagation without compensatory effects, using acute pharmacological inhibition. Backpropagation failures were not observed, consistent with results from conditional *Scn2a* knockout (**Spratt et al., 2021**). However, we did observe a reduction in the velocity at which the AIS component of the AP depolarized when Na$_V$1.2 was blocked selectively (**Figure 3—figure supplement 1B, C**). This suggests that proximal AIS-localized Na$_V$1.2 does aid in boosting backpropagating APs as they transit from the distal AIS to the soma. Nevertheless, Na$_V$1.2 channels are still dispensable for recruitment of somatic Na$_V$ channels, at least in mouse mPFC layer 5b pyramidal cells. Whether similar effects are observed in other cells, including those with larger somata that may be more difficult to depolarize in the absence of Na$_V$1.2, remains to be tested.

Na$_V$1.2 expression in pyramidal cell dendrites appears to have two roles in neuronal excitability. Intuitively, these channels provide local inward current that boosts dendritic excitability, leading to a recruitment of voltage-gated calcium channels (**Spratt et al., 2019**; **Nelson et al., 2024**). But this depolarization also appears to be critical for the recruitment of dendritically localized voltage-gated potassium channels that contribute to AP repolarization and net membrane potential between spikes (**Spratt et al., 2021**). Indeed, conditional knockout of Na$_V$1.2 paradoxically increases AP output, and we suggested previously that this was due in large part to loss of interactions between dendritic Na$_V$ channels and K$_V$ channels. But given observed changes in Na$_V$1.6 function as well as potential for other compensatory changes in non-Na$_V$ ion channels, it was difficult to ascribe hyperexcitability purely to the loss of Na$_V$1.2 alone (**Miralles and Patel, 2022**). Here, we showed that identical increases in excitability—associated with depolarization of membrane potential between APs—could be observed with acute Na$_V$1.2 block. This indicates that such effects can be due purely to the interplay between dendritic Na$_V$1.2 and K$_V$ channels, and that modifications to potassium channel distribution or function are not necessary for such effects.

Similar to Na$_V$1.2 conditional knockout, where Na$_V$1.6 is upregulated, knockout of Na$_V$1.6 results in an increase in Na$_V$1.2 expression, at least in the AIS. In pyramidal cells from mice constitutively lacking Na$_V$1.6, Na$_V$1.2 occupies the entirety of the AIS rather than just the region proximal to the soma (**Ye et al., 2018**; **Katz et al., 2018**; **Royeck et al., 2008**). Thus, it has not been possible to evaluate the role of Na$_V$1.6 in its normal distribution using genetic manipulations. Here, we find that acute Na$_V$1.6 block reduces AP output with a concomitant increase in AP threshold. This contrasted markedly with block of Na$_V$1.2, which had no effect on threshold. Of note, this distinction can be leveraged to assess

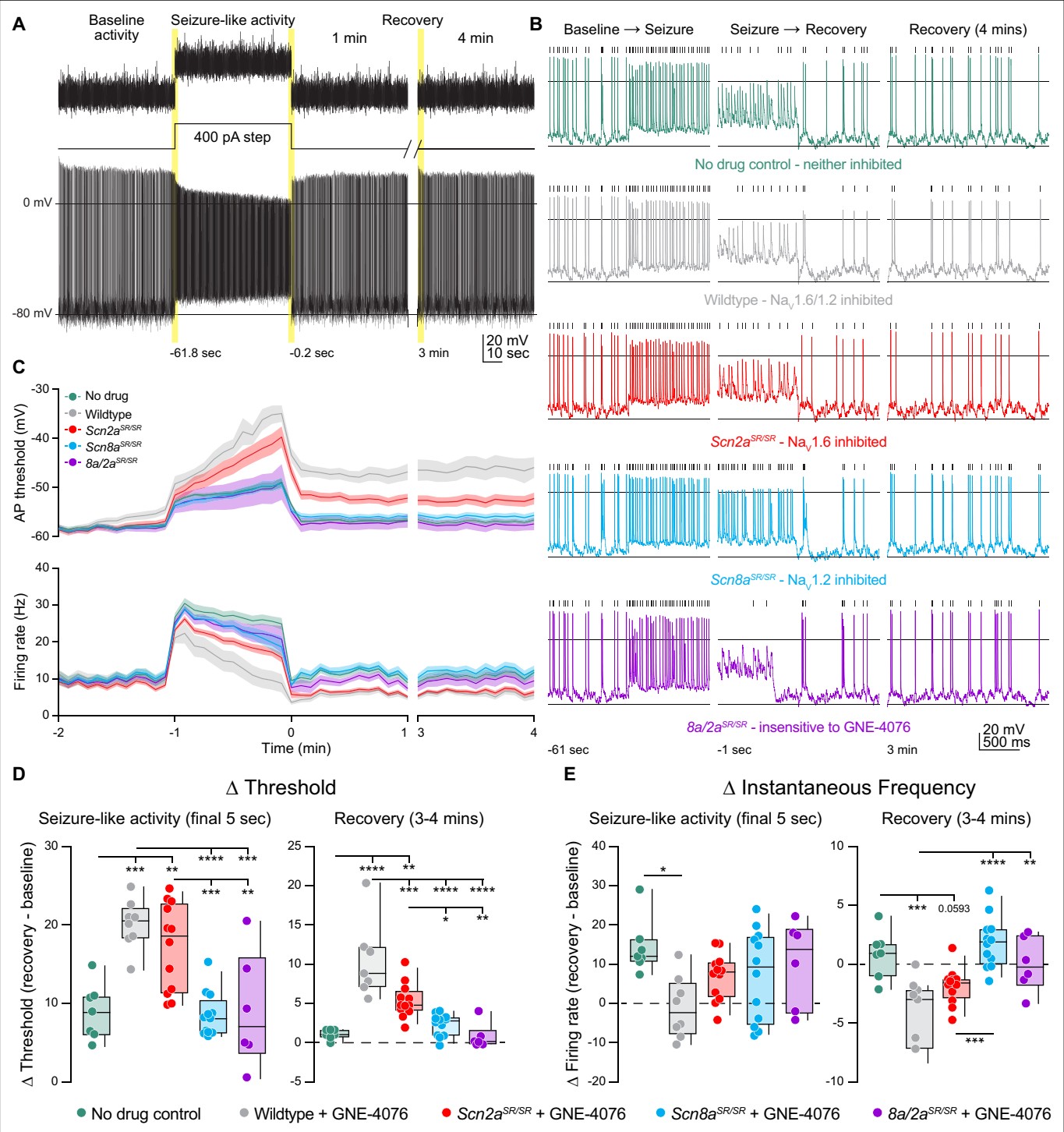

**Figure 7.** GNE-4076 onboarding following seizure-like activity continually impacts neuronal firing into recovery. (**A**) Stimulation protocol and example firing trace of cell injected with fluctuating post-synaptic potentials (PSPs) randomly generated using a Poisson probability distribution function for 60 s. PSPs were continuously applied to acquire baseline activity, seizure-like activity, and recovery activity. During seizure-like activity, a 400-pA step was applied in addition to the PSP. Recovery was continuously recorded for up to 4 min post seizure-like activity. (**B**) Zoomed-in example traces for all genotypes at the Baseline → Seizure transition, Seizure → Recovery transition, and start of 3–4 min recovery period (highlighted in (**A**)). Solid horizontal black bar represents membrane potential ($V_m$) of –12 mV. Tick marks above traces represent detected spikes defined as a change in $V_m$ of 15 V/s or greater. (**C**) Threshold (mV) or instantaneous firing frequency (Hz) binned in 5-s increments for all genotypes in (**B**). Solid lines represent mean value ± SEM. Timescale on x-axis mirrors activity presented in (**A**). (**D**) Summary of threshold data for the final 5 s of seizure-like activity or entire 3–4 min recovery time point in (**C**). Delta values are normalized to baseline activity (either at final 5 s or entire period). Box plots show median and 90% tails.

*Figure 7 continued on next page*

*Figure 7 continued*

Circles represent individual cells (wildtype no drug, $n = 7$; wildtype + GNE-4076, $n = 7$–8; $Scn2a^{SR/SR}$ + GNE-4076, $n = 12$; $Scn8a^{SR/SR}$ + GNE-4076, $n = 12$; $Scn8a/2a^{SR/SR}$ + GNE-4076, $n = 6$). One-way ANOVA, Holm–Šídák multiple comparisons test. *$p < 0.05$, **$p < 0.01$, ***$p < 0.001$, ****$p < 0.0001$. (**E**) Summary of instantaneous frequency data for the final 5 s of seizure-like activity or entire 3–4 min recovery time point in (**C**). Delta values are normalized to baseline activity (either at final 5 s or entire period). Box plots show median and 90% tails. Circles represent individual cells. One-way ANOVA, Holm–Šídák multiple comparisons test. *$p < 0.05$, **$p < 0.01$, ***$p < 0.001$, ****$p < 0.0001$.

The online version of this article includes the following source data and figure supplement(s) for figure 7:

**Source data 1.** Data values for each experiment described in *Figure 7*.

**Figure supplement 1.** GNE-4076 onboarding during baseline post-synaptic potential (PSP) activity.

**Figure supplement 1—source data 1.** Data values for each experiment described in *Figure 7-figure supplement 1*.

the specificity of other $Na_V$-targeting pharmacology that has been suggested to have specificity at select isoforms, as any change in threshold indicates that drugs are interacting with $Na_V1.6$ (*Ye et al., 2018*; *Filipis et al., 2023*).

## ASC pharmacology for suppression of neuronal hyperexcitability

Epilepsy can arise from genetic and non-genetic factors, often with unexplained etiology (*Lindy et al., 2018*; *Deng et al., 2014*). Seizure onset is broadly classified as an electrical imbalance of cellular and network activity that favors hyperexcitability (*Agbo et al., 2023*), whether it be cell intrinsic, synaptic, or due to complex network effects (*Greenfield, 2013*; *Yin et al., 2013*). Thus, seizure suppression can be targeted at multiple levels. Nevertheless, proper dosing can be difficult, as one aims to provide drug concentrations that temper excess activity but limit side effects like sedation associated with elevated drug concentrations.

For genetically defined sodium channelopathies, attention has been directed to neuronal cell types that express each isoform (e.g., *SCN1A*: $Na_V1.1$, *SCN2A*: $Na_V1.2$, and *SCN8A*: $Na_V1.6$) (*Encinas et al., 2020*; *Menezes et al., 2020*). Seizures resulting from *SCN1A* loss of function limit inhibitory neuron excitability, thereby disinhibiting excitatory pyramidal cells (*Favero et al., 2018*; *Tran et al., 2020*; *Strzelczyk and Schubert-Bast, 2022*). *SCN2A* gain-of-function results in hyperexcitability in pyramidal cells, especially in early development when $Na_V1.2$ channels are the sole isoform expressed in the AIS (*Jenkins and Bender, 2025*; *Gazina et al., 2015*). Later in development, *SCN8A*-encoded $Na_V1.6$ channels are expressed more ubiquitously in the AIS of most cell classes, supporting AP initiation in both excitatory and inhibitory cells (*Favero et al., 2018*; *Lopez-Santiago et al., 2017*; *Akin et al., 2015*; *Boiko et al., 2003*).

With these cellular distributions and mechanisms of action, guidelines have emerged for treatment. For *SCN1A* loss-of-function, sodium channel blocking anti-epileptics are typically contraindicated (*Kalume et al., 2013*; *Oakley et al., 2013*). This is because currently prescribed sodium channel blockers are non-selective, with little preference for $Na_V1.1$, 1.2, or 1.6 (*Bai et al., 2022*). Thus, further blocking of the remaining $Na_V1.1$ leads to even more disinhibition of excitatory cells before any beneficial effects of blocking $Na_V$ channels in excitatory cells are realized. Similarly, nonspecific sodium channel blockers are contraindicated for *SCN2A* loss-of-function seizures, as they tend to increase seizure severity. Instead, non-selective $Na_V$ inhibitors are useful when channel gain-of-function affects excitatory cells, as is the case for both *SCN2A* and *SCN8A* gain-of-function cases.

ASCs may be useful within each of these domains, as their chemistry can be adjusted to bias binding to specific isoforms in an activity-dependent manner (*Ahuja et al., 2015*; *Goodchild et al., 2024*; *Johnson et al., 2024*; *Johnson et al., 2022*). Indeed, compounds similar to those used here that inhibit both $Na_V1.2$ and $Na_V1.6$ (but not other $Na_V$ channels) are effective at suppressing chemoconvulsant-induced seizures in ex vivo models (*Goodchild et al., 2024*). This parallels our observations, where excess activity was dampened most effectively by dual block of $Na_V1.2$ and $Na_V1.6$ (*Figure 7C*). This effect appears due in large part to the block of $Na_V1.6$, since, ultimately, this is the isoform responsible for AP initiation. Consistent with this, $Na_V1.6$-preferring ASCs show promise in protecting from chemoconvulsant-induced seizures in vivo (*Johnson et al., 2024*; *Johnson et al., 2022*). Similar to our results, Johnson et al. also show that a selective ASC for $Na_V1.6$ (NBI-921352) preferentially targets cortical pyramidal cells, unlike carbamazepine that non-selectively targets many $Na_V$ isoforms leading to impaired

interneuron activity (*Johnson et al., 2022*). In fact, drug therapies using agents like carbamazepine, lamotrigine, and phenytoin that have little to no selectivity for $Na_V$ isoforms can have adverse effects in certain situations (*Agbo et al., 2023*). Thus, it may be that preferential block of $Na_V1.6$ would be beneficial in multiple sodium channelopathy conditions, including those associated with channel loss, since network hyperexcitability is ultimately dictated by $Na_V1.6$ function in the AIS of excitatory neurons after the first months of life.

Beyond selectivity, there may also be advantages to ASC activity-dependent properties. The ASC-binding pocket is hidden from drug in a channel's closed state. This feature, combined with on- and off-rate kinetics, results in an accumulation of block primarily for highly active neurons (*Figures 3 and 7*). Similar to on-demand, closed-loop electrical or optogenetic approaches for seizure intervention (*Krook-Magnuson et al., 2013*; *Nagaraj et al., 2015*; *Ledri et al., 2023*), ASCs are essentially biased to suppress prolonged, high-frequency activity, including the type of activity commonly observed during seizures. In this study, our focus centered mainly on seizure-like activity across individual neurons. Future studies can extend this approach to examining networks ex vivo and systems in vivo.

## Acknowledgements

We thank Drs. JP Johnson, Natali Minassian, Fiona Scott, and members of the Bender Lab for extensive discussions related to this work. This work was supported by NIH grants K00 MH134674 (JDG), MH125978 and MH126960 (KJB), by the Hartwell Foundation through an Individual Biomedical Research Award (RBS) and by FamiliesSCN2A through an Action Potential award (RBS).

## Additional information

### Competing interests

Emmie Banks, David H Hackos: Is an employee of Genentech. The other authors declare that no competing interests exist.

### Funding

| Funder | Grant reference number | Author |
| --- | --- | --- |
| National Institute of Mental Health | MH134674 | Joshua D Garcia |
| National Institute of Mental Health | MH125978 | Kevin J Bender |
| National Institute of Mental Health | MH126960 | Kevin J Bender |
| Hartwell Foundation | | Roy Ben-Shalom |
| FamiliesSCN2A Foundation | | Roy Ben-Shalom |

The funders had no role in study design, data collection, and interpretation, or the decision to submit the work for publication.

### Author contributions

Joshua D Garcia, Conceptualization, Formal analysis, Funding acquisition, Investigation, Visualization, Methodology, Writing – original draft, Writing – review and editing; Chenyu Wang, Ryan PD Alexander, Emmie Banks, Jean-Marc DeKeyser, Tatiana V Abramova, Formal analysis, Investigation, Methodology; Timothy Fenton, Software, Formal analysis, Investigation, Methodology, Writing – original draft; Alfred L George Jr, Conceptualization, Resources, Supervision, Writing – original draft, Project administration, Writing – review and editing; Roy Ben-Shalom, Data curation, Software, Formal analysis, Funding acquisition, Visualization, Writing – original draft, Writing – review and editing; David H Hackos, Conceptualization, Resources, Data curation, Formal analysis, Methodology, Writing – original draft, Project administration, Writing – review and editing; Kevin J Bender, Conceptualization,

Resources, Data curation, Formal analysis, Supervision, Funding acquisition, Validation, Investigation, Visualization, Methodology, Writing – original draft, Project administration, Writing – review and editing

### Author ORCIDs
Joshua D Garcia ⬤ https://orcid.org/0000-0002-7572-2062
Alfred L George Jr, ⬤ https://orcid.org/0000-0002-3993-966X
Kevin J Bender ⬤ https://orcid.org/0000-0001-7084-1532

### Ethics
This study was performed in strict accordance with the recommendations in the Guide for the Care and Use of Laboratory Animals of the National Institutes of Health. All animal procedures were in accordance with the Institutional Animal Care and Use Committee (IACUC) guidelines in accordance with the University of California, San Francisco (UCSF) and was approved by UCSF (protocol # AN194060).

Reviewer #1 (Public review): https://doi.org/10.7554/eLife.105696.3.sa1
Reviewer #2 (Public review): https://doi.org/10.7554/eLife.105696.3.sa2
Author response https://doi.org/10.7554/eLife.105696.3.sa3

---

## Additional files

### Supplementary files
MDAR checklist

### Data availability
Figure source data files contain the numerical data used to generate the figures.

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
