## [Editor Report · eLife Assessment]

This manuscript presents a clever and powerful approach to examining differential roles of Nav1.2 and Nav1.6 channels in excitability of neocortical pyramidal neurons, by engineering mice in which a sulfonamide inhibitor of both channels has reduced affinity for one or the other channels. Overall, the results in the manuscript are **compelling** and give **important** information about differential roles of Nav1.6 and Nav1.2 channels. Activity-dependent inactivation of NaV1.6 was also found to attenuate seizure-like activity in cells, demonstrating the promise of activity-dependent NaV1.6-specific pharmacotherapy for epilepsy.

---

## [Referee Report · Reviewer #1 (Public review)]

Summary:

Prior research indicates that NaV1.2 and NaV1.6 have different compartmental distributions, expression timelines in development, and roles in neuron function. The lack of subtype-specific tools to control Nav1.2 and Nav1.6 activity however has hampered efforts to define the role of each channel in neuronal behavior. The authors attempt to address the problem of subtype specificity here by using aryl sulfonamides (ASCs) to stabilize channels in the inactivated state in combination with mice carrying a mutation that renders NaV1.2 and/or NaV1.6 genetically resistant to the drug. Using this innovative approach, the authors find that action potential initiation is controlled by NaV1.6 while both NaV1.2 and NaV1.6 are involved in back-propagation of the action potential to the soma, corroborating previous findings. Additionally, NaV1.2 inhibition paradoxically increases firing rate, as has also been observed in genetic knockout models. Finally, the potential anticonvulsant properties of ASCs were tested. NaV1.6 inhibition but not NaV1.2 inhibition was found to decrease action potential firing in prefrontal cortex layer 5b pyramidal neurons in response to current injections designed to mimic inputs during seizure. This result is consistent with studies of loss-of-function Nav1.6 models and knockdown studies showing that these animals are resistant to certain seizure types. These results lend further support for the therapeutic promise of activity-dependent, NaV1.6-selective, inhibitors for epilepsy.

Strengths:

(1) The chemogenetic approaches used to achieve selective inhibition of NaV1.2 and NaV1.6 are innovative and help to resolve long-standing questions regarding the role of Nav1.2 and Nav1.6 in neuronal electrogenesis.

(2) The experimental design is overall rigorous, with appropriate controls included.

(3) The assays to elucidate the effects of channel inactivation on typical and seizure-like activity were well selected.

Weaknesses:

(1) As discussed in the revised manuscript, the fact that channels are only partially blocked by the ASC and that ASCs act in a use-dependent manner complicates the interpretation of the effects of NaV1.2 versus NaV1.6 on neuronal activity.

(2) The idea that use-dependent VGSC-acting drugs may be effective antiseizure medications is well established. Additional discussion of the existing, widely used, use-dependent VGSC drugs (e.g. Carbamazepine, Lamotrigine, Phenytoin) would improve the manuscript. Also, the idea that targeting NaV1.6 may be effective for seizures is established by studies using genetic models, knockdown, and partially selective pharmacology (e.g. NBI-921352). Additional discussion of how the results reported here are consistent with or differ from studies using these alternative approaches would improve the discussion.

---

## [Referee Report · Reviewer #2 (Public review)]

The authors used a clever and powerful approach to explore how Nav1.2 and Nav1.6 channels, which are both present in neocortical pyramidal neurons, differentially control firing properties of the neurons. Overall, the approach worked very well, and the results show very interesting differences when one or the other channel is partially inhibited. The experimental data is solid and the experimental data is very nicely complemented by a computational model incorporating the different localization of the two types of sodium channels.

The revised manuscript has re-organized figures that make the results and interpretation easier to follow.

---

## [Author Response]

The following is the authors’ response to the original reviews

**Public Reviews:**

**Reviewer #1 (Public review):**
Summary:Prior research indicates that NaV1.2 and NaV1.6 have different compartmental distributions, expression timelines in development, and roles in neuron function. The lack of subtype-specific tools to control Nav1.2 and Nav1.6 activity however has hampered efforts to define the role of each channel in neuronal behavior. The authors attempt to address the problem of subtype specificity here by using aryl sulfonamides (ASCs) to stabilize channels in the inactivated state in combination with mice carrying a mutation that renders NaV1.2 and/or NaV1.6 genetically resistant to the drug. Using this innovative approach, the authors find that action potential initiation is controlled by NaV1.6 while both NaV1.2 and NaV1.6 are involved in backpropagation of the action potential to the soma, corroborating previous findings. Additionally, NaV1.2 inhibition paradoxically increases the firing rate, as has also been observed in genetic knockout models. Finally, the potential anticonvulsant properties of ASCs were tested. NaV1.6 inhibition but not NaV1.2 inhibition was found to decrease action potential firing in prefrontal cortex layer 5b pyramidal neurons in response to current injections designed to mimic inputs during seizure. This result is consistent with studies of loss-of-function Nav1.6 models and knockdown studies showing that these animals are resistant to certain seizure types. These results lend further support for the therapeutic promise of activity-dependent, NaV1.6-selective, inhibitors for epilepsy.Strengths:(1) The chemogenetic approaches used to achieve selective inhibition of NaV1.2 and NaV1.6 are innovative and help resolve long-standing questions regarding the role of Nav1.2 and Nav1.6 in neuronal electrogenesis.(2) The experimental design is overall rigorous, with appropriate controls included.(3) The assays to elucidate the effects of channel inactivation on typical and seizure-like activity were well selected.Weaknesses:(1) The potential impact of the YW->SR mutation in the voltage sensor does not appear to have been sufficiently assessed. The activation/inactivation curves in Figure 1E show differences in both activation and inactivation at physiologically relevant membrane voltages, which may be significant even though the V1/2 and slope factors are roughly similar.

We have performed new experiments testing how YW->SR mutations affect spiking on their own. The reviewer’s intuition was correct; the small changes in voltage-dependence in NaV1.6 identified in heterologous expression systems translated into a ~2 mV hyperpolarization in threshold in neurons.

(2) Additional discussion of the fact that channels are only partially blocked by the ASC and that ASCs act in a use-dependent manner would improve the manuscript and help readers interpret these results.

We have updated text extensively to address this concern. Details are found in the author suggestions below.

(3) NaV1.6 was described as being exclusively responsible for the change in action potential threshold, but when NaV1.6 alone was inactivated, the effect was significantly reduced from the condition in which both channels were inactivated (Figure 4E). Similarly, Figure 6C shows that blockade of both channels causes threshold depolarization prior to the seizure-like event, but selective inactivation of NaV1.6 does not. As NaV1.2 does not appear to be involved in action potential initiation and threshold change, what is the mechanism of this dissimilarity between the NaV1.6 inactivation and combined NaV1.6/ NaV1.2 inactivation?

We believe the dissimilarity is due to interactions between NaV1.2 and other channel classes (e.g., potassium channels) throughout the cell, including the somatodendritic domain. NaV1.6 that initiates APs, localized to the AIS, do not live in isolation, and AP threshold can be affected by the recent membrane potential history. Loss of NaV1.2-mediated depolarization in the dendrites begets less potassium channel-mediated repolarization, as described in Figure 4.

(4) The idea that use-dependent VGSC-acting drugs may be effective antiseizure medications is well established. Additional discussion or at least acknowledgement of the existing, widely used, use-dependent VGSC drugs should be included (e.g. Carbamazepine, Lamotrigine, Phenytoin). Also, the idea that targeting NaV1.6 may be effective for seizures is established by studies using genetic models, knockdown, and partially selective pharmacology (e.g. NBI-921352). Additional discussion of how the results reported here are consistent with or differ from studies using these alternative approaches would improve the discussion

We agree; the concept of use-dependent block as a means to treat seizure is not new, and we have updated the discussion to include commentary on other medications currently in use. What is new here is our ability to explore the role of NaV1.2 and NaV1.6 in electrogenesis with a level of drug selectivity that could not be achieved without the addition of the YW->SR mutations. This approach in itself will not be useful in the clinic, but it may help guide drug design in the future. One major interpretation of this work is that NaV1.6 block is more effective than NaV1.2 block in general, and may even be effective for non-SCN8A genetic conditions. This is indeed one of the reasons that we believe that drugs like NBI-921352, itself an aryl-sulfonamide, is being tested in seizure models.

**Reviewer #2 (Public review):**
The authors used a clever and powerful approach to explore how Nav1.2 and Nav1.6 channels, which are both present in neocortical pyramidal neurons, differentially control firing properties of the neurons. Overall, the approach worked very well, and the results show very interesting differences when one or the other channel is partially inhibited. The experimental data is solid and the experimental data is very nicely complemented by a computational model incorporating the different localization of the two types of sodium channels.In my opinion the presentation and interpretation of the results could be improved by a more thorough discussion of the fact that only incomplete inhibition of the channels can be achieved by the inhibitor under physiological recording conditions and I thought the paper could be easier to digest if the figures were re-organized. However, the key results are well-documented.

This is a concern raised by multiple reviewers, and we thank you all for your help in improving the way in which we discuss the results. We have revised the manuscript extensively, moving figures around per your advice and the advice of R1 in their comments to authors.

**Reviewer #3 (Public review):**
Summary:The authors used powerful and novel reagents to carefully assess the roles of the voltage gated sodium channel (NaV) isoforms in regulating the neural excitability of principal neurons of the cerebral cortex. Using this approach, they were able to confirm that two different isoforms, NaV1.2 and NaV1.6 have distinct roles in electrogenesis of neocortical pyramidal neurons.Strengths:Development of very powerful transgenic mice in which NaV1.2 and/or NaV1.6 were modified to be insensitive to ASCs, a particular class of NaV blocker. This allowed them to test for roles of the two isoforms in an acute setting, without concerns of genetic or functional compensation that might result from a NaV channel knockout.Careful biophysical analysis of ASC effects on different NaV isoforms.Extensive and rigorous analysis of electrogenesis - action potential production - under conditions of blockade of either NaV1.2 or NaV1 or both.Weaknesses:Some results are overstated in that the representative example records provided do not directly support the conclusions.

We have swapped out example records to better capture the median effect observed and to better capture our discussion of these results. Please see below, in recommendations for authors, for details.

Results from a computational model are provided to make predictions of outcomes, but the computational approach is highly underdeveloped.

Modeling has been elaborated upon extensively, with more detail in methods, a new sensitivity analysis supplemental figure, and a deposition into ModelDB. Please see below, in recommendations for authors, for details.

**Reviewer #1 (Recommendations for the authors):**
Regarding the concern about the potential impact of the YWàSR mutation: All results in Figures 2-6 report only within-subject changes before and after drug-activating protocols. These results show that the drug has no effect on the mutant channel, but whether the mutant channel itself has any effect on neuronal properties is not clear. This deficiency could be rectified by reporting raw values for AP threshold, spike rate, etc. in the pre-drug condition and statistically analyzing the apparent differences in the activation/inactivation curves.

Data in our original submission only included data in the presence of GNE-4076. We now present new data showing how the YWàSR mutation affects baseline activity of neurons. These data are in Supplemental Figure 1. Compared to wildtype (no drug control) neurons, we observe no change in peak dV/dt. However, threshold is hyperpolarized by approximately 2 mV in dual knockin neurons (median values: -57.4 mV for dual knockin and -55 mV for wildtype). This is consistent with measures from heterologously expressed channels, where we observed somewhat subtle shifts in voltage-dependence of inactivation and activation in NaV1.6 as a result of YWàSR incorporation.

In addition to these data, we also include the baseline dataset from Figure 3, where GNE-4076 is present throughout recording, and report that neither threshold nor peak dV/dt are influenced by the presence of GNE at baseline. This suggests that any drug binding at baseline (i.e., before firing APs via somatic current injection) is negligible, consistent with the concept that GNE-4076 has low affinity for the closed channel state.

Minor Comments:While the single-cell response to "seizure-like" input aptly demonstrates the change in action potential threshold and firing rate induced by NaV1.6 inhibition, this component of the paper could be enhanced by a network-level assay that assesses the impact of this drug on an actual seizure-like event in acute slices or on seizure susceptibility in vivo.

This is an excellent thought, and the work near the end of this manuscript is an effort to mimic network-like activity in a controlled way in single cells. To expand this to bona fide seizure-like activity in acute slices or in vivo is something that we are considering for future studies. To do this properly requires extensive validation of dosing and seizure induction that will require several years’ effort.

Fig 1e caption says "circles" but the markers are squares

This has been corrected, thank you for catching it.

Color scheme in S2B is not intuitive to me

We’ve now updated the caption to better describe the color scheme used within.

Fig S2: graph or show change in threshold

Empirical threshold data are in main figure 3D. Changes in threshold related to modeling are now included in a new sensitivity analysis that is in a new Supplemental Figure 2.

Fig 3A example of NaV1.6 inhibition does not show change in AP threshold apparent in the aggregate data

We have updated the representative example to better illustrate the change in AP threshold for NaV1.6 inhibition.

"AP initiation is mediated exclusively by NaV1.6" not corroborated by data; APs still occur when NaV1.6 is inhibited

This was an over-interpretation of our data, indeed. We have updated the language to be more accurate to the following: “AP threshold and AP initiation appears to be initiated in an NaV1.6-rich region in control conditions; when NaV1.6 is inhibited, APs can occur at more depolarized potentials, likely mediated predominately by NaV1.2.”

Fig S3C missing WT/Scn8aSR/SR significance marking. Chosen example makes it look like there is a small decrease.

Please note that there is no difference between these two conditions when in delta dV/dt for AIS inflection point (p = 0.4344).

**Reviewer #2 (Recommendations for the authors):**
This manuscript presents a clever and powerful approach to examining differential roles of Nav1.2 and Nav1.6 channels in excitability of pyramidal cell excitability, by engineering mice in which a sulfonamide inhibitor of both channels has reduced affinity for one or the other. Overall, the results in the manuscript are interesting and give important information about differential roles of Nav1.6 and Nav1.2 channels.The paper makes an important contribution to better understanding distinct roles of Nav1.2 and Nav1.6 channels. This improved understanding could help guide design of anti-seizure drugs targeted to sodium channels.Having made it clear that I think this is an important and impressive piece of work for which the authors should be congratulated, I found reading and interpreting the manuscript a frustrating experience. I will be blunt about the ways in which I found the presentation and discussion to be frustrating and even annoying, in the spirit of frank feedback by one interested and appreciative reader that the authors can consider or reject as they wish.From the start, I had the feeling that the authors were presenting and discussing the results in a sanitized "never-mind-about the details" fashion such as might be appropriate for a seminar to a general audience not interested in details, but not appropriate for a research paper.

Our intent certainly was not to frustrate or annoy readers. We are very grateful that you have provided these comments, which have certainly improved the manuscript, hopefully mitigating some of the frustration for future readers. We appreciate that there are complex drug and voltage effects occurring within these studies, and in an effort to distill these effects into digestible prose, we appear to have been too earnest. We have expanded on the requested topics below and please note that, for the aficionados, every figure displays individual data. Further, we have made a special effort to ensure that features of excitability are presented throughout the drug and manipulation timecourse, including time-points before and after periods subject to statistical comparison, so that the reader may draw their own conclusions.

General:There were two major ways in which I found the presentation and discussion frustrating and even annoying: First, not clearly discussing early in the presentation the fact that it is impossible to achieve complete inhibition with this agent during measurements of physiological firing and second, presenting so much of the effects as deltas of various parameters rather than showing effects on absolute values of the parameters.

Our response to the first issue will follow the next comment, as it relates to this statement. Regarding use of deltas and absolute values for changes in threshold and dV/dt across figures. Every cell has a unique AP threshold and peak dV/dt, and we found that displaying data zeroed to baseline values best illustrated the effects of GNE-4076. Without this, GNE-based effect could be buried within the cell-to-cell variability. This helped most when trying to make the case that threshold was unaffected in 2a/8a YWàSR knockin animals. We continue to believe that this is the best way to display the data in the primary figures, but to provide a more complete account, we now present absolute values in supplemental tables and supplemental figures.

The first issue, the incomplete inhibition by the agent, was the most annoying because the authors obviously thought a lot about this and even closed the paper by proposing this as a positive feature of this class of inhibitors, yet discussed it only piecemeal - and with most of the key experimental data in the Supplement. There are two fundamental characteristics of this (and other) sulfonamide inhibitors that complicate interpretation of experiments, especially when applied in a slice experiment: they only bind to the channel when the channel is depolarized, and even when the channel is depolarized for many seconds, bind very slowly to the channel.That makes it almost impossible to know exactly what fraction of channels is being inhibited during measurements of firing. Obviously, the authors are well-aware of this issue and they allude to it and even make use of it in some of the protocols, but they never really discuss it in a very clear manner.

We agree that it is impossible to know the precise fraction of channels inhibited in acute slice preparations. But the reason for this is likely different than what has been interpreted by this reviewer. To state that ASMs “only bind to the channel when the channel is depolarized, and even when the channel is depolarized for many seconds, bind very slowly to the channel.” is not consistent with prior data on ASM–channel interactions. Clarification on these points may help the reviewer and a broader audience better understand the effects occurring here, and we appreciate being able to both address this concept here and by revising the manuscript.

First, ASMs bind activated channels and stabilize the inactivated state. It is correct that channels are more likely to enter these states when subject to voltage depolarization, but channel state is stochastic and can enter activated states near resting membrane potentials. The on-rate is fast enough that channels are blocked immediately in recordings in heterologous systems (Figure 1C). It is more likely that channel biophysical state stochasticity, along with drug concentration used herein, are likely dictating the rate at which channels accumulate block during repetitive spiking.

To address this in text, we have revised the 3rd paragraph of the introduction to better incorporate these ideas. This also helps with comments in the reviewer paragraph below.

The key experimental data on this is relegated to the Supplemental Figures. When the reader is first shown results of the effects of the inhibitor on firing in Fig 2, the presentation has been set up as if everything is perfect, and the inhibitor will be completely inhibiting either both or only one channel according to the mouse. With this presentation, it is then exceptionally striking that the cell in the middle panel of Fig 2A, labeled "Nav1.2/1.6 Inhibited" is firing action potentials very nicely even with both channels "inhibited". For a reader not already aware that there is likely only partial inhibition of each channel, the reaction will be "Huh? Shouldn't blocking both channels simply completely block excitability?". The authors do preface Fig 2 by a very brief allusion to the incomplete inhibition: "In spiking neurons, ASCs would therefore be predicted to exhibit use-dependence, progressively blocking channels in proportion to a neuron's activity rate" but this comes out of nowhere after the over-simplified picture of complete inhibition up to that point, and without any estimation of how much inhibition there is likely to be before activity, or how much induction of inhibition there is likely to be during the activity. Without this, interpreting the data in Fig 2 is basically impossible.The key experimental data on this issue is really in Supplemental Figures 1-2 and Fig 4, and I found myself immediately ping-ponging back and forth between the Supplemental figures and the main text trying to understand what is going on with the partial inhibition. This was frustrating.

Thank you for these suggestions; they help with readability appreciably. We have re-organized the figures presented in the manuscript and emphasized details about ASCs to ensure readers can discern between near-complete blockade of channels (Figures 1-4) and activity-dependent ASC onboarding (Figures 5-7). We now present near-complete block experiments *first*, detailing the current clamp-> voltage clamp (-12 mV)-> current clamp experiments. We incorporated Supp. Fig. 1 into main Figure 1 and moved Supp. Fig. 2 into main Fig. 2.

As the reviewer notes, there are clear time-dependent effects on channel function when stepping to -12 mV, independent of GNE-4076 block. As stated previously, “We therefore focused on the 12-20 sec after voltage-clamp offset for subsequent analysis, as it is a period in which most channel-intrinsic recovery has occurred, but also a period in which we would still expect significant block from GNE-4076.” We hope that reordering the manuscript as suggested and placing these results near the beginning will help with discerning between near-complete block and activity depending onboarding. By beginning with these experiments, which underscore that 100% block cannot be studied without “contamination” from native slow inactivation, we hope that the readers can better understand why data was done as presented.

In my opinion, the paper would be greatly improved by a detailed discussion of the voltage- and time-dependence of the inhibitor at the very beginning of the paper. For me, reading and digesting the paper would have been far easier if Fig 1 included a discussion of the voltage- and time-dependence of inhibition, and next Figs were then Supplemental Figs 1-2, and main Fig 4. The key questions are: how much inhibition is there before a 10-s current injection from the resting potential, and how much additional inhibition is there produced during either the 10-s bout of firing or the "on-boarding" depolarization protocol, and how long does that additional inhibition last? The most direct information on that is in the plots in Fig. 4D and Fig 4F in combination with Supplemental Fig 1, which shows that the on-boarding depolarization reduces current to about 30% of current before on-boarding. This is so central to the interpretation of all the results that I think Supp Fig 1 should be in the main paper as the first piece of data in neurons.

We originally had the nucleated patch data in supplement due to space constraints in an already large figure 1. Based on your recommendation we have moved it to the main figure. We have also changed the ordering of the paper and related figures to present data as suggested. Hopefully this better guides readers through the questions you are raising above, which are addressed in the (now reordered) figures mentioned above.

Specific:(1) Fig.1 I can find no information on the voltage protocol used to generate the dose-response curves. In the literature characterizing sulfonamide blockers, most protocols use very unphysiological strong, long depolarization to induce inhibition, usually with equally unphysiological short hyperpolarizations to produce recovery from inactivation. One assumes something like that was used here. Obviously, the protocol needs to be explained.

We updated the methods section to better describe the voltage protocol used to generate the dose response curves. In contrast to the literature characterizing sulfonamide blockers, we used pulses that closely mimic physiological activation from -80 mV (rest) to 0 mV (depolarized) for 20 msec. GNE-4076 was perfused onto cells at increasing concentrations throughout the experiment. At each successive dose, cells were held at 0 mV to allow adequate GNE-4076 onboarding.

(2) Supp Fig1. This shows the effect of depolarization to enhance inhibition, but not how much inhibition there was before the depolarization. Presumably, there were measurements during the application of drug? How much inhibition is there before the depolarization? Why does the time only go to 20-s, when the times in Figs 4 go to 10 minutes?

Nucleated patch recordings are notoriously difficult to maintain for long durations, especially when subjecting the patch to large voltage deflections. These recordings extend to 20s recovery periods because that is the duration for which we maintained all recordings, though some exhibited rather impressive longevity and allowed for several minutes of recording thereafter. Regardless, the goal here was to assess block within the 12-20 sec recovery window we utilized in current clamp recordings from intact neurons. This was achieved.

Please note that GNE-4076 was present throughout all recordings. This was in part due to time constraints, as we could not maintain patches long enough to also perform wash-in. The degree of inhibition can be inferred by comparing peak dV/dt and threshold of cells in the absence and presence of GNE-4076. These data are presented in a new Supplemental figure 1, showing no difference in threshold or peak dV/dt.

(3) Fig. 4. Similar question here - this is a very nice and informative figure, but we see only the delta in threshold and dv/dt, but how were the initial absolute values different in the drug compared to control?

These data are presented in a new Supplemental Figure 1, showing no difference in threshold or peak dV/dt.

(4) Fig 2. As far as I can tell, we have no idea how much inhibition there is at rest, before the current injection -what is the dv/dt in the drug compared to in the control? Were there experiments in which the current injections were delivered before and after applying drug? If not, at least it would be useful to see population data on dv/dt of the first spike in control and with drug.

These data are presented in a new Supplemental Figure 1, showing no difference in threshold or peak dV/dt.

(5). Fig. 2. Do the authors have any quantitative information on how much extra inhibition would be produced at 200 nM drug using physiological waveforms of firing?

These types of analyses are part of later figures using EPSC-like waveforms to evoke spiking.

I was unconvinced that the changes in threshold and dv/dt during the firing in the drug necessarily represent time-dependent use-dependent effects of drug. Partial inhibition by TTX would probably produce greater progressive changes in spike shape and reduced ability to fire robustly.

TTX is not use-dependent, so it is a good contrast to GNE-4076. We experimented with a few cells at 2 and 10 nM TTX concentrations and found that concentrations required to mimic the block of spiking that occurs with 200 nM GNE-4076 in WT cells was associated with a marked use-independent elevation in AP threshold, with an inability to maintain ~10 Hz spiking rates with the baseline EPSC-like stimulation pattern. These effects are very different from those produced by GNE-4076, but were expected given the use-independence of TTX. We did not pursue this line of inquiry fully, so we present these data only as individual examples in the reviewer figure below:

**Author response image 1. sa3fig1:** Data from Figure 6B, D, E are replicated here with individual lines of 2 nM and 10 nM TTX shown in dashed lines. Note marked changes in threshold not observed with GNE-4076. TTX sourced from Alomone Labs.

Minor:p. 5 and elsewhere: it seems unnecessary to give values of threshold and dv/dt to three decimal places, especially when the precision is not better than a single decimal place.

We have reduced unnecessary precision throughout.

**Reviewer #3 (Recommendations for the authors):**
The computational model is highly underdeveloped. Without more rigorous development the results of the computational model appear to provides little additional insight beyond that expected from the known axodendritic localizations of NaV 1.2 and 1.6. If the authors wish to use the computational results to make rigorous predictions, then this section needs to be either be expanded to be more complete and promoted to a regular figure, with full details of the model, and how it was evaluated for accuracy. Alternatively, this point regarding computational insight could be de-emphasized and or removed from the paper.Modeling:(1) I don't see any methods describing the precise model parameters that were used.

Apologies, this is a model that we have built and tested extensively over the years (PMID: 38290518, 35417922, 34348157, 31995133, 31230762, 28256214), though there have been some small updates over these works. We have deposited this model at ModelDB and provide data there regarding model construction (access #2019342).

(2) There appears to be no robustness test to assess whether the particular results/conclusions were unduly dependent on particular model construction decisions.

We have now generated a new supplemental figure 2 that explores the robustness of these observations to changes in NaV1.2 and NaV1.6 position within the AIS and changes in relative density of NaV1.2 and NaV1.6. As shown there, the model is tolerant to all but extreme, non-physiological manipulations to these parameters.

(3) Figure S2 does not really provide convincing evidence of a biologically relevant model. Probably the model itself needs to be redesigned to better replicate the biological response and be validated by testing parameter sensitivity.a) All of the results in S2C show that there is a huge reduction in the first action potential (black?) followed by relatively little change in subsequent spikes. This is not seen in any of the models. The progressive changes in threshold as predicted by the model for dual and NaV1.6 block are not at all evident in the results of C, except perhaps for the the very first and the very last spikes.b) The baseline action potential in B is different than the recorded action potentials. In particular, the somatic depolarization occurs much later and over a more extended time frame than the real neuron, and the phase plot shows an actual dip in depolarization at the transition to the somatic spike, which is not representative of naturally occurring action potentials.

To address both (a) and (b), please note that in empirical experiments there are two parallel processes occurring: block by GNE-4076 and channel recovery from inactivation. In the model we can isolate the effects of block to test that parameter fully and in isolation. This is something that we could never achieve biologically. The important take home here in both cases is to observe that with NaV1.6 block there is a change in threshold, whereas with NaV1.2 block there is none.

(4) The one finding that seems to be robust is that the changes in NaV1.2 have little effect on threshold.

Yes! This is a major take-home message from both the model and the use of these knockin mice in combination with GNE-4076. In mature pyramidal cells, NaV1.6 is the major determinant of AP threshold. And to editorialize on this observation, changes in threshold are a useful metric to test if other pharmacology are truly selective for NaV1.2 over NaV1.6. We note that phrixotoxin-3, which is described as NaV1.2 specific in multiple papers, was never tested for specificity over NaV1.6 in its original description, and we find that it fails this test in our hands.

Data presentation:(1) The phase plots in Figure 3B (left and right) appear to be visually identical, and as such don't strongly support any particular conclusion.

We changed the representative example record (specifically for Fig. 3A-B) to more directly support the conclusions.

(2) It is unclear to me what is meant by AP speed (title of Figure 3 legend). Do the authors mean propagation speed along the axon, or perhaps the rate of action potential firing?

Apologies, we are referencing dV/dt when we mention AP speed. We updated AP speed to AP velocity throughout the manuscript.